# Arctic temperature and precipitation extremes in present-day and future storyline-based variable resolution Community Earth System Model simulations

René R. Wijngaard<sup>1,4</sup>, Willem Jan van de Berg<sup>1</sup>, Christiaan T. van Dalum<sup>1,5</sup>, Adam R. Herrington<sup>2</sup>, and Xavier J. Levine<sup>3,6</sup>

Correspondence: René R. Wijngaard (r.r.wijngaard@uu.nl)

**Abstract.** Over the last few decades, the Arctic region has warmed up at a greater rate than elsewhere on the globe, partly resulting from the on-going loss of sea ice and seasonal snow over land. It is projected that the amplified warming of the surface will continue in the future. In addition, the intensity and frequency of temperature and precipitation means and extremes are projected to change, which may pose serious threats for human infrastructure and livelihoods. To assess (future) climate extremes, advanced modelling approaches with (regionally) refined resolution could be helpful.

In this study, we use the variable-resolution Community Earth System Model version 2.2 (VR-CESM) to evaluate and assess present-day and future temperature and precipitation extremes, such as heat waves and heavy precipitation, over the Arctic. Applying a globally uniform 1° grid and a VR grid with regional grid refinements to 28 km over the Arctic and Antarctica, we run 30-year present-day (1985–2014), 10-year present-day (2005–2014), and future (2090–2099) simulations with interactive atmosphere and land surface models, and prescribed sea ice and sea surface temperatures. We use the 30year simulation to evaluate the ability of the VR grid to simulate climate extremes by comparison with gridded outputs of the globally uniform 1° grid, reanalysis-based datasets, and a regional climate model. The 10-year simulations follow two storylines of Arctic climate change representing a combination of strong/weak Arctic tropospheric warming and weak/strong sea surface warming in the Barents-Kara Seas and are used to assess future climate extremes by focussing on temperature and precipitation extremes. The outcomes show that the VR grid generally performs better in simulating precipitation extremes, while the globally uniform 1° grid generally performs better in simulating temperature extremes, which is mainly related to larger negative temperature differences in the VR grid. Future projections suggest that high temperature extremes will generally increase both in intensity and duration, whereas low temperature extremes will decrease in intensity and duration, especially over regions dominated by sea surface warming and large sea ice loss. Further, wet precipitation extremes are projected to increase in intensity and frequency. The outcomes of this study may contribute to an improved understanding on future climate extremes and its implications.

<sup>&</sup>lt;sup>1</sup>Institute for Marine and Atmosphere Research Utrecht, Utrecht University, Utrecht, the Netherlands

<sup>&</sup>lt;sup>2</sup>Climate and Global Dynamics Laboratory, NSF National Center for Atmospheric Research, Boulder, CO, USA

<sup>&</sup>lt;sup>3</sup>NORCE Norwegian Research Centre, Bjerknes Centre for Climate Research, Bergen, Norway

<sup>&</sup>lt;sup>4</sup>Now at: Department of Physical Geography, Utrecht University, Utrecht, the Netherlands

<sup>&</sup>lt;sup>5</sup>Now at: Royal Netherlands Meteorological Institute, De Bilt, the Netherlands

<sup>&</sup>lt;sup>6</sup>Now at: Columbia University, New York, USA

#### 1 Introduction

In recent decades, the Arctic region has warmed up at a greater rate than elsewhere on the globe, a phenomenon known as "Arctic amplification" (Rantanen et al., 2022). Although the exact causes are still under debate, numerous studies have shown that Arctic amplification is caused by several feedback mechanisms involving interactions between the atmosphere, ocean, sea ice, and land surface, such as enhanced ocean-atmosphere coupling and changing surface albedos resulting from the ongoing loss of sea ice and snow over land (Screen and Simmonds, 2010; Dai et al., 2019; Park et al., 2019; Previdi et al., 2021; Rantanen et al., 2022). Under increasingly warmer climate conditions, high temperature extremes, such as warm days/nights and heat waves, have become more frequent in recent decades, while low temperature extremes, such as cold days/nights and cold spells, have become less frequent (Matthes et al., 2015; Walsh et al., 2020). In contrast to temperature extremes, systematic trends for precipitation extremes have not been observed and are more regional in nature due to the greater spatial variability of precipitation, the lack of measurements at high latitudes and altitudes, the accuracy of measurements, and uncertainties associated with atmospheric reanalyses (Walsh et al., 2020). According to Walsh et al. (2020), increasing trends in precipitation extremes have been observed over much of the Arctic land area, although regionally decreasing trends or no trends have also been observed.

In the future, Arctic warming is projected to continue, altering the intensity and frequency of temperature extremes (Sillmann et al., 2013b; Screen et al., 2015; Landrum and Holland, 2020; Walsh et al., 2020). As warming continues, precipitation is expected to increase, generally in line with the Clausius-Clapeyron relationship, which describes an increase in atmospheric moisture content under warmer climate conditions (Pfahl et al., 2017). In addition, precipitation is expected to increasingly shift from snow to rain, and the intensity and frequency of wet precipitation extremes are projected to increase, while those of dry precipitation extremes are projected to decrease (Sillmann et al., 2013b; Screen et al., 2015; Landrum and Holland, 2020; Walsh et al., 2020; Paik et al., 2023). The changing intensity and frequency of temperature and precipitation extremes could ultimately lead to more frequent floods, wildfires, and reduced agricultural production, with profound impacts on ecosystems, human infrastructure and livelihoods (Hirabayashi et al., 2013; Masrur et al., 2018; Walsh et al., 2020; Overland, 2022). Given the potential future impacts of temperature and precipitation extremes in the Arctic, there is a need to develop adaptation and mitigation strategies that can help reduce potential adverse impacts on vulnerable Arctic communities and ecosystems.

Developing future adaptation and mitigation strategies is a challenging task due to the wide range of climate change projections resulting from uncertainties in possible future greenhouse gas emission scenarios, incomplete understanding of physical processes and their representation in climate models, and natural variability within the climate system (Hawkins and Sutton, 2009; Overland et al., 2019; McCrystall et al., 2021; Levine et al., 2024). To strengthen decision-making processes related to the development of adaptation and mitigation strategies, a number of possible climate outcomes – storylines – can be investigated. Storylines can be described as a physically self-consistent unfolding of past events or a plausible future pathway representative for regional climate change (Shepherd et al., 2018). Storylines have been generated using two distinct methodologies accord-

ing to the goal they want to achieve. Event-based storylines use an extreme synoptic event and apply changes to the mean state of the atmosphere to quantify the resulting change in impacts (e.g., Sillmann et al., 2021; Chan et al., 2022). Dynamical storylines use a multi-variate linear regression to generate climate states based upon the dependence of those climate state to predetermined climate indices; those climate indices generally represent a well-known change in the atmospheric, oceanic or sea ice state (e.g., Zappa and Shepherd, 2017; Zappa, 2019; Mindlin et al., 2020; Williams et al., 2024). Recently, dynamical storylines for the Arctic have been developed by Levine et al. (2024) that describe future pathways for Arctic summer climate change. These storylines are based on two different drivers that explain substantial fractions of the surface climate response to global warming in the Arctic, namely: 1) warming of the Arctic lower troposphere and 2) warming of the sea surface in the Barents-Kara Sea. As both drivers are expected to lead to changes in the occurrence and intensity of temperature and precipitation extremes, the Arctic storylines could contribute to a better understanding of the possible range of impacts of regional climate change on the intensity and frequency of temperature and precipitation extremes over the Arctic.

To investigate present-day and future climate extremes in the Arctic, a variety of advanced modelling approaches have been used. Global climate models (GCMs) have often been used to assess the present-day state and future changes in temperature and precipitation extremes on a global scale (e.g., Sillmann et al., 2013b, a; Seneviratne and Hauser, 2020; Kim et al., 2020; Seneviratne et al., 2021). Although GCMs simulate temperature extremes and large-scale precipitation extremes reasonably well, high-resolution models with horizontal grid spacings of 0.25° or higher have shown the ability to simulate precipitation extremes better than coarser-gridded models (Wehner et al., 2014; O'Brien et al., 2016; Seneviratne et al., 2021). However, the use of high-resolution GCMs has been limited due to the large computational resources that are required to run these GCMs. In this context, regional climate models (RCMs) could be considered as a more suitable alternative. RCMs, such as those used in the Arctic-CORDEX experiment (https://climate-cryosphere.org/arctic-cordex/), can be run at a higher spatio-temporal resolution with horizontal grid spacings as fine as ~11 km, and are therefore able to better capture spatio-temporal variability in temperature and precipitation. In addition, RCMs are often more specialized than GCMs in the simulation of (polar) processes (e.g. the treatment of snow and ice) and more optimized for specific regions of interest. Nonetheless, RCMs need to be forced with GCMs or reanalysis products, which disables two-way interactions between the region of interest and the global domain and introduces inconsistencies in terms of model physics and dynamics between RCMs and GCMs.

To overcome the limitations associated with RCMs and the computational constraints associated with high-resolution GCMs, variable-resolution GCMs or Earth System Models (ESMs), such as the variable-resolution Community Earth System Model (VR-CESM), have been developed. VR-CESM (Zarzycki et al., 2014; Lauritzen et al., 2018) is a hybrid between regional and global climate models as it applies a regional grid refinement over a region of interest within a coarse-gridded global domain (Rhoades et al., 2016). VR-CESM has been used to study various processes in several regions. For example, with regional refinements up to 7 km, VR-CESM has been used to study regional climate, atmospheric rivers, wind extremes, glacier surface mass balance, and/or snowpack characteristics in the western USA (Rhoades et al., 2018), Chilean Andes (Bambach et al., 2021), Mediterranean (Boza et al., 2025), East Asia (Zhu et al., 2023), High Mountain Asia (Wijngaard et al., 2023), Greenland (van Kampenhout et al., 2019; Herrington et al., 2022; Loeb et al., 2024; Waling et al., 2024), and Antarctica (Datta et al., 2023). Furthermore, present-day and future climatic means and/or extremes have been assessed with VR-CESM over

western US, Canada, eastern China, and Greenland (Xu et al., 2022; Yin et al., 2024; Morris et al., 2023, 2024; Morris and Kushner, 2025). However, assessments focusing on temperature and precipitation extremes over the entire Arctic region have not been conducted thus far.

The main objective of this study is to evaluate and assess present-day and future temperature and precipitation extremes over the Arctic, using VR-CESM. To this end, we apply a globally uniform 1° (~111 km) grid and a dual polar VR grid with horizontally refined grid spacings of 28 km (0.25°) over the Arctic and Antarctic with interactively coupled atmosphere and land surface models, and prescribed sea ice and sea surface temperatures. We run three different types of simulations with both model grids, namely: 1) present-day simulations covering a 30-year period (1985–2014), 2) present-day simulations covering a 10-year period (2005–2014), and 3) future simulations (2090–2099) following the high-emission SSP5-8.5 scenario. The 30-year present-day simulations are used to evaluate the performance of the VR-CESM grids through comparisons with gridded outputs from reanalysis datasets and a regional climate model. The 10-year present-day and future simulations are used to assess future changes in temperature and precipitation extremes by following two storylines of Arctic summer climate change representing a combination of strong/weak Arctic tropospheric warming and weak/strong sea surface warming in the Barents-Kara Seas. Compared to previous VR-CESM studies focusing on climate extremes and/or the Arctic, this study has several novelties. First, this study is the first VR-CESM application that evaluates and assesses present-day and future intensity and occurrence of temperature and precipitation extremes over the entire Arctic. Second, we use a storyline approach to assess future extremes over the Arctic, which can potentially improve decision-making processes related to the development of adaptation and mitigation strategies.

This paper is organized as follows. Section 2 briefly describes the model and highlights the methods and data. Section 3 presents and discusses the main outcomes of this study. Finally, Section 4 provides further discussion and the conclusions.

#### 110 2 Data and Methods

100

105

# 2.1 Modelling setup

We used the Community Earth System Model version 2.2 (CESM2; Danabasoglu et al., 2020; Herrington et al., 2022), a state-of-the-art global Earth system model consisting of multiple model components (including atmosphere, oceans, land surface, rivers, sea ice and land ice) that can be run in a partially or fully coupled mode. In this study, we applied CESM in a partially coupled mode by interactively coupling prognostic atmosphere and land surface components, and using prescribed daily sea ice and sea surface temperatures to replace the active ocean and sea ice components. This model configuration follows the Atmospheric Model Intercomparison Project (AMIP) protocol (Gates et al., 1999).

The atmosphere component of CESM2, the Community Atmosphere Model version 6 (CAM6), is applied with a hydrostatic spectral element dynamical core that supports unstructured grids that eliminates polar singularities and enables VR capabilities (CAM6-SE; Zarzycki et al., 2014; Lauritzen et al., 2018; Gettelman et al., 2019). Physics parameterization schemes applied in CAM6 include the Cloud Layers Unified by Binormals (CLUBB) scheme simulating shallow convection, boundary layer turbulence, and cloud macrophysics (Bogenschutz et al., 2013); a deep convection scheme (Zhang and McFarlane, 1995); a

two-moment cloud microphysics scheme with prognostic treatment of precipitation (MG2; Gettelman and Morrison, 2015); the modal aerosol module (MAM4; Liu et al., 2016); a radiation scheme (Rapid Radiative Transfer Method for GCMs - RRTMG; Iacono et al., 2008); and anisotropic orographic gravity wave (Weimer et al., 2023) and form drag parameterization schemes (Beljaars et al., 2004). Our atmospheric model configuration uses a dry mass low-top vertical coordinate with a model top at ~40 km and 58 hybrid sigma-pressure levels in the vertical instead of the standard 32 levels of CESM2 (Lauritzen et al., 2018). The new enhanced vertical grid is planned for the CESM3 model and has a higher resolution in the planetary boundary layer and in the middle and upper troposphere, and is intended to improve the representation of moisture, temperature, and cloud profiles in the boundary layer and to reduce noise and spurious flow features (Skamarock et al., 2019; Huang et al., 2022). CAM6 is coupled to the land surface component of CESM2, the Community Land Model version 5 (CLM5; Lawrence et al., 2019), which is applied with satellite vegetation phenology (CLM5-SP). CLM5 simulates the surface energy balance, hydrological processes, biogeochemical cycles, and their interactions with the atmosphere (Oleson et al., 2013). It includes several new and updated processes and parameterizations, such as for snow and surface discretization (Lawrence et al., 2019).

#### 2.1.1 Grids

The VR-CESM simulations performed in this study are run with two different spectral element grids: 1) a globally uniform  $1^{\circ}$  ( $\sim$ 111 km) grid, hereafter referred to as NE30, and 2) a dual polar VR grid (Fig. 1), hereafter referred to as POLARRES. The POLARRES grid was generated by the SQuadGen software package (Ullrich, 2014). Partly based on the existing ARCTIC and ANTSI VR grids (Herrington et al., 2022; Datta et al., 2023), POLARRES has horizontal grid spacings of  $0.25^{\circ}$  ( $\sim$ 28 km) over the Arctic and Antarctic (poleward of  $60^{\circ}$ N/S). Further, a transition grid with horizontal grid spacings of  $0.5^{\circ}$  ( $\sim$ 55 km; poleward of  $45^{\circ}$ N/S) serves as a buffer between the Arctic and Antarctic domains and the global  $1^{\circ}$  ( $\sim$ 111 km) domain.

To ensure numerical stability, CAM physics (dynamics) time steps of 1800 s (300 s) and 450 s (75 s) were used for NE30 and POLARRES, respectively. The VR-CESM simulations were performed on the Cheyenne supercomputing facility at the NSF National Center for Atmospheric Research (NSF-NCAR; Computational and Laboratory, 2019). The computational cost of the NE30 and POLARRES grids amount to about 5 300 and 50 000 core hours per simulated year, respectively.

The topography of the VR-CESM grids was interpolated from an updated 30-arcsec global topography dataset comprising of the Global Multi-resolution Terrain Elevation Data (GMTED2010; Danielson and Gesch, 2011) of the United States Geological Survey (USGS) and the BedMachine topography (Morlighem et al., 2017, 2020), which includes terrain elevations of Greenland and Antarctica (Wijngaard et al., 2023). To interpolate the topography, we used the NSF-NCAR topography generation software package (Lauritzen et al., 2015).

# 2.1.2 Parameter tunings

The regionally refined resolution associated with the POLARRES simulations most likely has an impact on the representation of clouds as found in previous studies performed with VR-CESM (Wijngaard et al., 2023; Boza et al., 2025). For this reason, we implemented additional parameter tunings (largely following Wijngaard et al. (2023)) for the POLARRES simulations to tune the low-level cloudiness, and the shortwave radiation and albedo over sea ice. First, we increased the strength of the

damping for the 3rd moment of the vertical velocity in the large skewness regime  $(clubb\_c11b)$  from 0.35 to 0.375. Increasing the damping strength reduces the vertical velocity skewness of the cloud distribution and increases the fraction of low cloud cover (Guo et al., 2015; Boza et al., 2025). Second, to tune shortwave radiation and albedo over sea ice, we decreased the snowmelt onset temperature  $(dt\_melt)$  over sea ice from 1.5 to 1.0 °C, and decreased the snow grain radius by increasing the parameter  $(r \ snw)$  over sea ice from 1.25 to 1.5 standard deviations, resulting in a higher snow albedo.

In addition, we implemented modifications to fix a model bug in the cloud microphysics (Shaw et al., 2022; Zhu et al., 2022). These modifications are largely based on the paleoclimate-calibrated CESM2 configuration of Zhu et al. (2022), which include a reduction of the microphysical time-step size by increasing the number of microphysical substeps, and the removal of a limiter on the cloud ice number concentrations. For the NE30 simulations, we have adopted the number of microphysical substeps (micro\_mg\_num\_steps) from Zhu et al. (2022), which increases from 1 (default value) to 8, resulting in a microphysical time-step size of 75 s. For the POLARRES simulations, we maintained the microphysical time-step size of 75 s, which is derived by increasing the number of microphysical substeps from 1 to 3.

## 2.2 Experimental Design

180

To evaluate and assess present-day and future climate extremes, three different model experiments are run with and without regional grid refinement over the Arctic. First, we run present-day experiments for a 30-year period (1985–2014) while forcing VR-CESM with daily sea ice cover (SIC) and sea surface temperatures (SST) retrieved from the ERA5 reanalysis dataset (Hersbach et al., 2020). These experiments (hereafter referred to as present-day A experiments) were used to evaluate the performance of the model grids in simulating present-day climate extremes and to understand the impact of regional grid refinement on the representation of climate extremes. Second, we run future storyline experiments for the period 2090–2099 using SST and SIC fields from CMIP6 models representing selected Arctic storylines that are physically plausible outcomes of Arctic summer climate change. The selected storylines follow the high-emission SSP5-8.5 scenario and were derived by Levine et al. (2024) based on two different climate drivers: 1) Arctic atmospheric warming at the 850 hPa level and 2) sea surface warming in the Barents-Kara Seas. In this study, we used two different Arctic storylines that show opposite paths of Arctic climate change, namely:

- 1. Storyline ST1 (corresponding to storyline D in Levine et al. (2024)): Strong Arctic tropospheric warming combined with weak sea surface warming in the Barents-Kara Seas (ArcAmp+BKWarm-). Relative to the multi-model mean (MMM) climate change, this storyline is characterized by warmer and drier continents, weaker sea surface warming over the Arctic and North Atlantic Ocean (Fig. 2c), and reduced sea ice loss (Fig. 2g).
- 2. Storyline ST2 (corresponding to storyline A in Levine et al. (2024)): Weak Arctic tropospheric warming combined with strong sea surface warming in the Barents-Kara Seas (ArcAmp-BKWarm+). Relative to the multi-model mean (MMM) climate change, this storyline is characterized by cooler and wetter continents, stronger sea surface warming (Fig. 2d), and stronger sea ice loss, especially in the Barents-Kara Seas (Fig. 2h).

Each storyline is represented by a CMIP6 model selected for its similarity to the storylines and its skill to simulate the historical climate. Here, the r1i1p1f1 member of the Norwegian Earth System Model version 2 (NorESM2-MM; Seland et al., 2020) and the r1i1p1f2 member of the CNRM-CERFACS Earth System Model version 2 (CNRM-ESM2-1; Séférian et al., 2019) model realizations represent storylines ST1 and ST2, respectively. Finally, we run another set of present-day experiments for a 10-year period (2005–2014) using SST and SIC fields from the same CMIP6 model realizations used for the future storyline experiments. These baseline experiments (hereafter referred to as present-day B experiments) are used (in combination with the future storyline experiments) to assess future changes in climate extremes over the Arctic. The CMIP6 model realizations representing ST1 and ST2 project a global mean near-surface warming of 3.3 K for ST1 and 4.6 K for ST2 between the future and present-day periods. All model experiments were spun-up for a period of 1 year.

Greenhouse gas concentrations, tropospheric aerosols, and ozone concentrations are time-varying and prescribed in accordance with CMIP6 historical and SSP5-8.5 forcings for the present-day and future storyline experiments, respectively. Stratospheric aerosols were not prescribed for the model experiments in this study. The land surface distributions for the VR-CESM grids are partly transient and partly constant. The plant and crop functional type distributions are transient and are interpolated from the Land Use Harmonization (LUH2) time series, which has been developed for CMIP6 (Hurtt et al., 2020). The distributions of other land surface classes, such as glaciers, lakes, and urban areas, are assumed to be constant in time and set to the distributions of the year 2000.

#### 2.3 Model evaluation

- To evaluate the performance of the NE30 and POLARRES grids in simulating the climatology and extremes of temperature and precipitation, we compared gridded outputs from the present-day A experiments (1985–2014) with gridded outputs from two different reanalysis products and an RCM. To this end, we used daily precipitation sums and daily maximum, minimum, and mean 2-meter temperature from ERA5 (reanalysis), JRA-3Q (reanalysis), and RACMO (RCM):
  - The ERA5 reanalysis data (Hersbach et al., 2020) were retrieved from the KNMI Climate Explorer (https://climexp. knmi.nl/start.cgi; last access: 2 July 2024). Originally, the ERA5 reanalysis data are available on a ∼31 km grid and monthly or hourly intervals. However, the data retrieved from the KNMI Climate Explorer have been regridded to a spatial resolution of 0.5°x0.5° and are available at daily intervals.
    - The Japanese Reanalysis for Three Quarters of a Century (JRA-3Q) reanalysis (Kosaka et al., 2024) is the successor of the JRA-55 reanalysis, which has recently been released by the Japanese Meteorological Agency. JRA-3Q reanalysis data have been retrieved from the Research Data Archive (RDA) of NSF-NCAR (https://rda.ucar.edu/datasets/ds640.0; last access: 12 August 2024) and are available on a TL479 grid (~40 km).
    - Regional Atmosphere Climate Model (RACMO) data are based on RACMO version 2.4p1, a recently updated RCM that
      is specially developed to simulate polar climate processes over Greenland, the Arctic, and Antarctica (van Dalum et al.,
      2024). RACMO is forced with SST, SIC, and multi-level ERA5 data of wind speed, temperature, humidity, and pressure,
      and is available on an 11 km grid.

Although reanalysis data are commonly used to evaluate model outcomes, there may be some limitations to its use. This is because reanalysis fields can be strongly influenced by the underlying forecast model in regions with little or no observations. To allow comparison between the CESM grids and reference data, all CESM output and reference data were regridded to a 1-degree finite volume grid (0.9°x1.25°), unless noted otherwise. Daily temperature fields were regridded using bilinear interpolation, and daily precipitation fields were regridded using conservative interpolation. The regridding was applied prior to the analysis of temperature and precipitation climatology and extremes. Finally, we have not used the high-resolution CARRA (Copernicus Arctic Regional Reanalysis) reanalysis due to its limited domain coverage. The release of a new version of CARRA, covering the entire Arctic domain, is planned for 2025–2026 (CARRA, 2025).

# 2.4 Analysis of temperature and precipitation extremes

To evaluate and assess present-day and future temperature and precipitation extremes in the Arctic, we selected eight extreme metrics (Table 1), including four temperature extreme metrics and four precipitation extreme metrics. These metrics were analyzed on a year-by-year basis using the Climate Data Operators (CDO) functions based on the European Climate Assessment (ECA) climate indices, which are consistent with the definitions of the Expert Team on Climate Change Detection and Indices (ETCCDI).

To analyze temperature extremes, we selected extreme metrics representing the intensity and duration of temperature extremes. The annual maximum of daily maximum temperature (TXx) and the annual minimum of daily minimum temperature (TNn) were used to analyze changes in the intensity of temperature extremes. To analyze changes in the duration of temperature extremes, we used the Warm Spell Duration Index (WSDI) and Cold Spell Duration Index (CSDI) as proxies for heat waves and cold spells, respectively. The warm spell duration index is defined as the annual number of days in intervals of at least 6 consecutive days on which the daily maximum temperature (TX) exceeds the  $90^{th}$  percentile of TX. The cold spell duration index refers to intervals during which the daily minimum temperature (TN) falls below the  $10^{th}$  percentile of TN. The  $90^{th}$  and  $10^{th}$  percentiles of daily maximum and minimum temperature, respectively, are calculated for each calendar day using a 15-day running window for the base periods (1985–2014 for present-day A and 2005–2014 for present-day B). To derive the future warm and cold spell duration metrics, we used the historical  $90^{th}$  and  $10^{th}$  percentiles, calculated for the present-day B base period, as a threshold.

We assessed the intensity of precipitation extremes by analyzing the  $99^{th}$  percentile of daily precipitation sums (P99) and the highest 5-day precipitation sums (RX5day). We also analyzed the number of heavy precipitation days (R10mm; defined as daily precipitation equal to or greater than  $10 \text{ mm d}^{-1}$ ) and the greatest number of consecutive dry days with daily precipitation less than  $1 \text{ mm d}^{-1}$  (CDD) to assess the frequency of heavy precipitation days throughout the year and the longest duration of dry spells, respectively.

The climatology and extremes of temperature and precipitation are analyzed by using probability density functions (PDFs) and regional averages that are calculated for four different Arctic land regions, one ocean region and one sea ice region covering areas poleward of 60°N (Fig. 1). Arctic land regions are based on regional domains defined by Seneviratne et al. (2012), namely Alaska/western Canada (AWC; 105-168°W; 60-75°N), eastern Canada/Greenland (ECG; 10-105°W; 60-85°N), Scandinavia

(SCA; 0-40°E; 60-85°N) and Siberia (SIB; 40-180°E; 60-85°N). The sea ice region (SIC) is based on the maximum extent of sea ice cover during the present-day A (1985–2014) or present-day B (2005–2014) periods, taking into account all grid cells with sea ice cover greater than 15%. The ocean region (OCN) includes all ocean grid cells poleward of 60°N that are not classified as sea ice. The PDFs are constructed by sampling temperature or precipitation for all grid points in a specific region, while regional averages are based on metrics that have been calculated for each grid point within the respective regions. To avoid the effects of polar singularities on PDFs, CESM output and reference data are regridded to the NE30 grid (to allow a uniform comparison between the CESM grids and reference data), prior to the construction of the PDFs.

To evaluate statistical significance, we performed two-tailed T-tests, where outcomes are marked as statistically significant when p-values are lower than or equal to 0.05. As the statistical testing is applied to a large number of grid points, there is an increased likelihood of false positives emerging from multiple statistical testing (Wilks, 2016). To control for these false positives, we used the False Discovery Rate (FDR) approach (Benjamini and Hochberg, 1995; Wilks, 2016) to correct the p-values. Here, we set the FDR control level ( $\alpha_{FDR}$ ) to 0.1, which corresponds to a global significance level of 0.05 (Wilks, 2016).

#### 3 Results

265

#### 3.1 Evaluation

## 270 3.1.1 Temperature Extremes

Before assessing future changes in temperature and precipitation extremes in the Arctic, we evaluate the performance of NE30 and POLARRES in representing temperature and precipitation climatology and extremes. Figure 3 shows the present-day A (1985–2014) annual mean near-surface temperature differences between the NE30 and POLARRES grids, and the gridded outputs of ERA5, JRA-3Q, and RACMO. NE30 shows higher temperatures primarily over Europe and parts of Greenland, while these differences are less significant in POLARRES. Furthermore, both NE30 and POLARRES show lower temperatures over the central Arctic, northeastern Canada, and parts of Greenland, where POLARRES is on average colder than NE30. The cold temperature differences are largest compared to ERA5, with average differences of -1.32 °C for NE30 and -1.72 °C for POLARRES (Figs. 3a-b). The differences are smaller compared to RACMO, with average differences of -1.24 °C for NE30 and -1.64 °C for POLARRES (Figs. 3e-f), and compared to JRA-3Q, with average differences of -0.54 °C and -0.95 °C, respectively (Figs. 3c-d).

Lower temperatures over the central Arctic, northeastern Canada, and Greenland are also visible in the present-day A daily temperature probability density functions (PDFs), as shown in Figure 4. These PDFs demonstrate that CESM is colder than ERA5 and RACMO over eastern Canada/Greenland (ECG) and the sea ice region (SIC), as well as being colder than JRA-3Q over the sea ice region (Figs. 4b,f). In particular, in low temperature regimes (i.e., the lower tail) it is 2-5 °C colder in NE30/POLARRES than in ERA5, JRA-3Q, or RACMO, with temperatures below -30 °C occurring more frequently in CESM and temperatures between -30 °C and -10 °C occurring more frequently in ERA5, JRA-3Q, or RACMO. In other regions, the

PDFs of CESM and the reference data are mostly similar, except for temperature regimes around 0°C, which tend to occur more frequently in CESM over the land regions (Figs. 4a-d).

To evaluate the performance of CESM in simulating temperature extremes, and to gain a better understanding of the patterns of the lower temperatures over the Arctic, we compare CESM output with ERA5. Figure 5 shows the differences between CESM and ERA5 for the present-day A temperature extreme metrics. For the annual maximum temperature, NE30 and PO-LARRES show lower temperatures over northeastern Canada and the central Arctic (Figs. 5b-c), and higher temperatures over parts of the continental mid-latitudes. In general, POLARRES shows small improvements in the annual maximum temperature with a small reduction in the temperature difference of 0.2 °C. Although POLARRES is generally warmer over Siberia. Alaska, and Scandinavia and colder over northeastern Canada, the differences between POLARRES and NE30 are mostly insignificant (Fig. 5d). For the warm spell duration index, the differences between the CESM grids and ERA5 are mostly negative but insignificant (Figs. 5f-g), with the differences becoming slightly more negative in POLARRES (Fig. 5h). The annual minimum temperature shows larger differences than the annual maximum temperature, with higher temperatures over land and, to a lesser extent, over the oceans, as well as large cold temperature differences over the central Arctic (Figs 5j-k). Here, the annual minimum temperature difference is slightly more negative, though still insignificant, in POLARRES, with a decrease in the annual minimum temperature over most of the Arctic, especially over Greenland, Iceland, Scandinavia, Svalbard and the Barents-Kara Sea region (Fig. 51). The decrease in the annual minimum temperature may be partly due to better resolved topography. Cold spell duration index differences are generally negative but insignificant over much of the central Arctic and Greenland in NE30 and over parts of Scandinavia in POLARRES (Figs. 5n-p), indicating a smaller number of cold spell days than in ERA5. Although the annual mean temperature and the daily minimum temperature both show negative temperature differences relative to ERA5 - particularly over the central Arctic and parts of Greenland (Figs. 3 and S1f-h) - the negative differences are not unexpected. One possible explanation for these differences is that the variability of the daily minimum temperature differs between ERA5 and CESM. For example, the variability of the daily minimum temperature over sea ice is lower in ERA5 than in CESM (not shown), which may be partly due to the fixed sea ice thickness in ERA5 (i.e., 1.5 m; Batrak and Müller, 2019). In CESM, however, sea ice thickness varies, leading to higher variability in the daily minimum temperature and thus a lower probability of meeting the cold spell criteria (Table 1). This ultimately results in a lower number of cold spell days.

Comparisons between CESM output and other reference data (i.e., JRA-3Q and RACMO) (Figs. S2 and S3) show broadly similar patterns, including lower annual maximum temperatures over northeastern Canada, lower annual minimum temperatures over the sea-ice-dominated regions, and higher annual minimum temperatures over most land areas (Figs. S2b-c,j-k and S3b-c,j-k). The annual maximum and minimum temperature differences generally correspond well to the climatological differences (Fig. 3). However, the magnitude of the extreme temperature differences is much larger. One possible explanation for the lower temperatures is the large warm near-surface temperature biases of 5-10°C over sea ice in reanalysis products, such as ERA5 and JRA-55 (the predecessor of JRA-3Q), during clear sky conditions in winter (Batrak and Müller, 2019; Zampieri et al., 2023). These biases are mainly caused by the poor representation of the snow layer on top of the sea ice and the thickness of the sea ice itself (Batrak and Müller, 2019; Zampieri et al., 2023). The warm biases in reanalysis products can also explain,

to some extent, the cold temperature differences between CESM and RACMO, since RACMO is forced with sea ice, SST, and multi-level data (temperature, wind, humidity, and pressure) from ERA5, and thus inherits ERA5 uncertainties. Uncertainties in reanalysis products, such as ERA5, could suggest that the lower temperatures in CESM may be exaggerated. To better understand the magnitude of these differences, validation of near-surface temperature against observations from meteorological stations in the Arctic is recommended in future research.

Another possible explanation for the lower temperatures in CESM is a bias in the cloud forcing. For instance, Herrington et al. (2022) identified negative biases in the summer shortwave cloud forcing across much of the Arctic, suggesting excessive reflection and cooling. Similar negative biases were also identified in NE30 and POLARRES over the Arctic land regions (Figs. S4a-c), particularly over northeastern Canada. However, these biases cannot fully explain the enhanced cold temperature differences over northeastern Canada in POLARRES. Although the summer shortwave cloud forcing bias is more positive in POLARRES, the near-surface temperature still shows cooling relative to NE30. This is possibly due to increased summer surface albedo over northeastern Canada and the central Arctic (Figs. S4d-f). The increased summer surface albedo may be related to the increased snow depth over northeastern Canada during the summer season (not shown). Finally, the higher temperatures over land in CESM are most likely related to a precipitation deficit, leading to a soil moisture deficit, and ultimately resulting in less evaporative cooling and higher temperatures (Boza et al., 2025; Lin et al., 2017; Lorenz et al., 2016).

### 3.1.2 Precipitation Extremes

350

355

Figure 6 shows the present-day A annual mean precipitation differences between the NE30 and POLARRES grids, and the gridded outputs from ERA5, JRA-3Q, and RACMO. CESM generally tends to be wetter than ERA5, JRA-3Q, or RACMO over the ocean and most of the Arctic land regions (i.e., poleward of 60°), and drier over most of the continental mid-latitudes, which, to some extent, supports the existence of a precipitation deficit that contributes to the higher near-surface temperatures over the continental mid-latitudes (Figs. 3 and 5). On average, CESM is wetter than ERA5, JRA-3Q, or RACMO with average differences ranging from 0.01-0.06 mm d<sup>-1</sup> in NE30 (Figs. 6a,c,e) to 0.02-0.07 mm d<sup>-1</sup> in POLARRES (Figs. 6b,d,e), with the smallest differences relative to RACMO (Figs. 6e-f), followed by JRA-3Q (Figs. 6c-d) and ERA5 (Figs. 6a-b). The wetter conditions in POLARRES are primarily related to the higher precipitation rates over elevated terrain, such as the mountainous terrains of Scandinavia, southern Iceland, and southern Alaska. This is mainly due to the better resolved topography.

Figure 7 shows the present-day A daily precipitation PDFs, which have been constructed on a NE30 grid to enable a uniform comparison between the CESM grids and the reference data (also see Section 2.4). Generally, extreme precipitation rates are lowest in NE30 and highest in RACMO, particularly over Alaska/western Canada, eastern Canada/Greenland, the ocean and the sea ice region (Figs. 7a-c,f). POLARRES, JRA-3Q and ERA5 often exhibit similar extreme precipitation rates. Only over Siberia, ERA5 and JRA-3Q, and, to lesser extent, RACMO do show lower extreme precipitation rates than NE30 and POLARRES (Fig. 7d). It is possible that extreme precipitation rates are underestimated in ERA5 and JRA-3Q, as, for example is found for ERA5 in Siberia (Clelland et al., 2024). The lower precipitation rates in RACMO could be attributed to uncertainties in the ERA5 data used to force RACMO. Over the ocean, the spread between CESM output and reference data is relatively small compared to the spread in other regions (Fig. 7e). The differences in extreme precipitation rates between the CESM

output and the reference data can generally be explained by differences in horizontal resolution. This is lowest for NE30 ( $\sim$ 111 km), similar for POLARRES ( $\sim$ 28 km), JRA-3Q ( $\sim$ 40 km) and ERA5 ( $\sim$ 31 km), and highest for RACMO (11 km). The increase in the frequency of extreme precipitation rates with increased horizontal resolution may, for instance, be related to 1) a better representation of topography, leading to stronger orographic uplifts and orographic precipitation, 2) stronger updrafts, resulting in higher resolved-scale precipitation rates, and 3) an increase in the frequency and intensity of storm systems (e.g., Zarzycki and Jablonowski, 2014; O'Brien et al., 2016; Xu et al., 2022).

To better understand the patterns of precipitation extremes over the Arctic, we compare the CESM output with ERA5. Figure 8 shows the differences between CESM and ERA5 for the present-day A precipitation extreme metrics. Compared to ERA5, the  $99^{th}$  percentile of daily precipitation over the continents is lower in NE30, particularly in mountainous regions, such as those along the eastern Pacific, Scandinavia, and southeastern Greenland and Iceland (Fig. 8b). In POLARRES, the 99<sup>th</sup> percentile of daily precipitation increases over the same regions, but particularly on the windward side of the mountainous regions, while it decreases on the more inland or leeward side, which is particularly evident over southeastern Greenland (Figs. 8c-d). The highest 5-day precipitation sum differences are positive over parts of eastern Asia, Greenland, and North America in NE30 (Fig. 8f). Dry precipitation differences are present over only a few regions, including parts of western Canada, Europe, and western Russia. In POLARRES, precipitation differences increase, particularly over parts of the Eurasian continent (Figs. 8g-h). The larger differences in POLARRES may be partly due to a higher frequency and intensity of storm systems, which bring significant amounts of precipitation over a period of several days (Zarzycki and Jablonowski, 2014). The number of heavy precipitation days shows positive differences over the northern Atlantic and negative differences over land in NE30, particularly over mountainous regions along the eastern Pacific, southeastern Greenland and Iceland, Scandinavia, and Scotland (Fig. 8j). In POLARRES, positive differences decrease over the North Atlantic (Figs. 8k-l). Furthermore, there is a significant increase in heavy precipitation days over southeastern Iceland, Scandinavia, and the eastern Pacific, and a decrease over the more inland or leeward sides of mountainous regions in southeastern Greenland, Scandinavia, and the eastern Pacific. Finally, the greatest number of consecutive dry days shows negative differences over parts of the central Arctic in NE30 and POLARRES, which could indicate a more variable precipitation climatology in CESM, reducing the length of dry periods (Figs. 8n-o). Relative to NE30, consecutive dry days increase slightly in POLARRES, however, differences between POLARRES and NE30 are mostly insignificant (Fig. 8p).

Comparisons between CESM output and other reference data (JRA-3Q and RACMO; Figs. S5 and S6) show that the spatial patterns are mostly similar compared to the differences between NE30/POLARRES and ERA5. Compared to JRA-3Q, CESM grids show a similar performance compared to ERA5, although the dry precipitation differences for CESM appears slightly larger. Compared to RACMO, the differences are generally much more negative, indicating that precipitation in RACMO is more extreme in intensity and frequency, which is consistent with the larger extreme precipitation rates shown in Figure 7. The differences in extreme precipitation rates and metrics could be related to the different horizontal resolutions, with higher resolution grids generally simulating larger extreme precipitation rates and metrics, which is consistent with other studies showing the effect of horizontal resolution on (extreme) precipitation (Herrington et al., 2022; Bacmeister et al., 2014; Huang et al., 2022; Xu et al., 2022), as well as to different physics and parameterizations. The higher agreement between POLARRES and

JRA-3Q and ERA5 suggests that POLARRES can better represent extreme precipitation than NE30. However, the discrepancies between POLARRES and RACMO also suggest to some extent that higher resolution grids are needed to further improve the representation of extreme precipitation, which is also demonstrated by CESM simulations on (storm-resolving) grids with a horizontal resolution of 14 km or higher (Huang et al., 2022; Xu et al., 2022).

# 395 3.2 Future projections

## 3.2.1 Temperature Extremes

Towards the end of the 21<sup>st</sup> century, both storylines project an increase in the magnitude of mean temperature and temperature extremes. Figure 9 shows the present-day B (2005–2014) annual mean near-surface temperature and the projected future (2090–2099) changes in annual mean near-surface temperature for two different storylines and CESM grids (NE30 and POLARRES), and the differences between the grids. The present-day panels (Figs. 9a,c,e,g) show that the annual mean temperature is relatively similar for storylines ST1 (associated with strong Arctic tropospheric warming) and ST2 (associated with strong sea surface warming in the Barents-Kara Seas), with ST1 being slightly colder than ST2 over the central Arctic. Additionally, the present-day B annual mean temperature is slightly lower than the present-day A annual mean temperature over the central Arctic (not shown), due to the lower SST originating from the CMIP6 models representing the storylines (NorESM2-MM for ST1 and CNRM-ESM2-1 for ST2), which force CESM.

Present-day differences between POLARRES and NE30 (Figs. 9i,k) indicate that POLARRES is generally colder than NE30, especially in ST1. Only over parts of Greenland, Alaska and eastern Siberia, POLARRES is warmer than NE30. These differences are partly due to the better resolved topography, particularly over Alaska, Greenland, Iceland, Scandinavia and parts of Siberia. Towards the end of the century, both storylines show a significant increase in temperature (Figs. 9b,d,f,h). ST2 is characterized by a warmer Arctic in the future, with temperature increases of up to about 19 °C over the Barents-Kara Seas (Figs. 9d,h), which can be linked to strong sea surface warming in the region (Fig. 2d). ST1 shows a local temperature decrease (up to about 2-3 °C) over the northern Atlantic (Figs. 9b,f), due to SST cold anomalies in the region (Fig. 2c). These cold anomalies are most likely related to increased freshwater inflow and the subsequent weakening of the Atlantic Meridional Overturning Circulation (AMOC) (Liu et al., 2020; Weijer et al., 2020). Future differences between POLARRES and NE30 (Figs. 9j,l) are relatively small and insignificant for both storylines.

Stronger ST2 temperature responses are also evident from the projected changes in regional temperature extremes as shown in Figure 10. In all regions, the annual maximum and minimum temperatures are projected to increase. The projected changes in high temperature extremes are smaller than the projected changes in annual mean near-surface temperature, while the projected changes in low temperature extremes are larger (Figs. 10a-b). Additionally, ST2 increases are considerably larger than ST1 increases, with average differences of up to about 9°C for annual minimum temperature between ST1 and ST2 (Figs. 10a-b). Specifically over the sea ice region, NE30 and POLARRES project temperature increases of up to about 9°C and 9°C for annual maximum temperature, and 19°C and 20°C for annual minimum temperature, respectively. This suggests that sea surface warming and associated sea ice loss have the strongest effect on temperature over the sea ice itself.

Temperature increases over land are generally smaller with increases in the range of 3-8°C and 6-9°C for the annual maximum temperature, and in the range of 7-11 °C and 12-14 °C for the annual minimum temperature in ST1 and ST2, respectively. Here, POLARRES generally predicts similar or larger increases in annual minimum temperature than NE30 and smaller annual maximum temperature increases over Alaska/western Canada and Siberia, as well as larger annual maximum temperature increases over eastern Canada/Greenland and Scandinavia. The differences over eastern Canada/Greenland are associated with significantly stronger warming over northeastern Canada in POLARRES (Fig. S7b), mostly due to large decreases in snow depth in this region (not shown). As the present-day snow depth is overestimated, this warming trend is likely to be spurious. The differences over other land regions, such as Scandinavia, are mostly related to either stronger or weaker, though insignificant, warming over land (Fig. S7d). Temperature increases over the ocean are relatively small compared to the changes over land and sea ice, especially for ST1. This is most likely due to the projected cooling over the northern Atlantic (Figs. S8b,j and S9b,j).

Consistent with the expected temperature increases, warm spell duration also increases, particularly under ST2 and over the sea ice region. Conversely, cold spell duration decreases in all regions (Note that we have derived the future warm and cold spell duration metrics based on the present-day B percentile thresholds; see Section 2.4). The increases in warm spell duration and decreases in cold spell duration under ST1 are smaller over the ocean and, to a lesser extent, over Scandinavia (for warm spell duration only) than in other regions. These smaller changes can be attributed to the projected cooling-induced changes in warm spell (decrease) and cold spell (increase) duration over the northern Atlantic (Figs. S8f,n and S9f,n).

Although the differences in warm spell duration between POLARRES and NE30 are relatively small, they are larger locally, e.g. over northeastern Canada or western Russia (Figs. S7f,h), where POLARRES projects a higher warm spell duration. The cold spell duration differences are generally less negative. The projected changes in warm spell (increase) and cold spell (decrease) duration are consistent with the findings of Screen et al. (2015), who assessed future changes in regional climate extremes arising from Arctic sea ice loss, using regions similar to those used in this study. Although the magnitude of change in cold spell duration is similar to that found in Screen et al. (2015), the projected changes in warm spell duration are much larger in this study. For example, in eastern Canada/Greenland, the projected increase in warm spell duration in the Screen et al. (2015) study is up to about 35 dyr<sup>-1</sup>, while in this study the projected increases are up to about 220 dyr<sup>-1</sup>. A possible explanation for these differences is that the future experiments performed in Screen et al. (2015) were based on "idealized" model experiments using prescribed present-day SST and future sea ice concentrations, whereas our experiments use both prescribed future SST and sea ice concentrations. This indicates that not only sea ice loss but also sea surface warming is an important driver of future changes in temperature extremes.

# 3.2.2 Precipitation Extremes

Figure 11 shows the present-day B (2005–2014) annual mean precipitation and projected future (2090–2099) changes in annual mean precipitation for two different storylines and CESM grids, and the differences between the grids. The present-day annual precipitation is similar for both ST1 and ST2 (Figs. 11a,c,e,g). However, regionally the annual mean precipitation is larger in POLARRES, particularly on the windward sides of mountainous regions in the eastern Pacific/southern Alaska, southeastern

Greenland, Iceland, and Scandinavia. In contrast, precipitation tends to be smaller on the leeward or more inland sides of these mountainous regions (Figs. 11i,k). In the future, both NE30 and POLARRES project an increase in annual precipitation over most of the Arctic in both storylines, with the exception of the north(east)ern Atlantic where precipitation is projected to decrease (Figs. 11b,f,d,h). These decreases are larger in ST1 and are associated with decreases in SST and near-surface temperature over the region (Figs. 2c-d and 9b,f,d,h). As the projected cooling is greater in ST1, the amount of atmospheric moisture available for precipitation (as depicted by the Clausius-Clapeyron relationship) is less and can therefore explain the greater precipitation decreases in ST1. Future differences between POLARRES and NE30 (Figs. 11j,l) show consistent regional patterns in both storylines with wetter but insignificant trends in POLARRES over the northern Atlantic, the Scandinavian mountain ranges, and the eastern Pacific coast, and drier but insignificant trends over Greenland and some parts of the land (e.g. eastern Siberia in ST1 and Alaska in ST2).

Figure 12 shows the projected changes in regional precipitation extremes for two different storylines and CESM grids. In ST1, the  $99^{th}$  percentile of daily precipitation, the highest 5-day precipitation sum, and the number of heavy precipitation days increase, on average, by about 3 mmd<sup>-1</sup>, 6-7 mm, and 3 dyr<sup>-1</sup>, respectively, in NE30 and POLARRES (i.e. calculated for the domain  $60-90^{\circ}$ N). In ST2, the projected increases are slightly larger with average increases of about 4 mmd<sup>-1</sup>, 8-9 mm, and 4 dyr<sup>-1</sup> for the  $99^{th}$  percentile of daily precipitation, the highest 5-day precipitation sum, and heavy precipitation days, respectively. These increases correspond to mean relative increases of about 25-26%, 29-30%, and 36-40% in ST1 and 30-31%, 38%, and 43-50% in ST2, respectively (Fig. S10b-d), indicating that the relative changes in heavy precipitation days are the largest. As the annual mean precipitation increases by  $\sim$ 0.3 mmd<sup>-1</sup> (20-21%) in ST1 and  $\sim$ 0.4 mmd<sup>-1</sup> (25-26%) in ST2 (Figs. 11b,f,d,h; Fig. S10a), the projected changes in extremes exceed the changes in the means. Regionally, the ST2 increases are often larger than the ST1 increases (Figs. 12a-c). Only over Alaska/western Canada the ST2 increases are slightly smaller than the ST1 increases for the  $99^{th}$  percentile of daily precipitation and the heavy precipitation days (Figs. 12a,c). The larger increases in ST2 are most likely related to the larger temperature increases in ST2, which lead to an increase in atmospheric moisture and thus more precipitation.

Consistent with the increase in mean and extreme precipitation, the number of consecutive dry days is projected to decrease in most regions, except for Scandinavia where it is expected to increase (Fig. 12d). Similar increases were also found in Scandinavia in Screen et al. (2015). Regions where consecutive dry days are projected to decrease are characterized by a high present-day number of consecutive dry days. These regions (central Arctic, north(east)ern Greenland and Canada, and eastern Siberia; Figs. S11m-p, S12m-p) are located at high latitudes and experience cold winters in which daily precipitation above 1 mm d<sup>-1</sup> is relatively rare. With increased warming, precipitation increases and consequently consecutive dry days reduce, suggesting that these regions become wetter. Regions with a lower present-day number of consecutive dry days are generally located in the warmer, lower-latitude parts of the Arctic, where more precipitation occurs. In a warmer climate, these regions are expected to become wetter, as mean precipitation and precipitation extremes are projected to increase in intensity and frequency. However, these regions may also experience longer periods of drought in summer in a warmer climate, resulting in an increased number of consecutive dry days. This results in a higher precipitation variability, particularly over regions such as Scandinavia.

Regarding the differences between NE30 and POLARRES for future precipitation (extremes), the differences often show mixed signals depending on the extreme metric and the region, although for some regions and extreme metrics the differences are more pronounced. For example, over Scandinavia POLARRES predicts larger absolute increases in the 99<sup>th</sup> percentile of daily precipitation, the highest 5-day precipitation sum, and heavy precipitation days for both storylines (Figs. 12 and S13). Furthermore, the highest 5-day precipitation sum differences are more pronounced over Alaska/western Canada, Siberia, and the sea ice region, where POLARRES predicts larger increases. The larger increases in the highest 5-day precipitation sums may be partly due to the higher frequency and intensity of storm systems associated with higher resolution grids, as discussed above (Fig. 8). The larger increases over Scandinavia may partly be explained by increased moisture fluxes and enhanced moisture convergence associated with grid refinement, as seen in Figure 13.

520

525

Figure 13 shows the future changes in the annual mean vertically integrated moisture flux convergence, and the 850 hPa zonal wind and moisture fluxes for ST1/ST2 and NE30/POLARRES, and the differences between NE30 and POLARRES. Both NE30 and POLARRES predict a strengthening of westerly moisture transport over most of the higher mid-latitudes (i.e. roughly between 50°N and 70°N) in ST1 and over North America, northern Atlantic, and northern Europe in ST2 (Figs. 13ac,g-i and S14). The NE30 moisture flux anomalies are associated with an eastward shift of the low-level North Atlantic jet in ST1 (Fig. 13d). The eastward shift of the North Atlantic jet also occurs in POLARRES but is accompanied by enhanced zonal wind near the climatological maximum of the jet (Figs. 13e-f). Here, the zonal wind changes are mainly due to changes occurring during winter (Figs. S15d-f and S16d-f). In ST2, the projected changes in zonal wind are much weaker compared to those in ST1, where NE30 does not show significant changes in zonal wind over the northern Atlantic region (Fig. 13j). However, in ST2, POLARRES predicts a poleward shift of the jet and a significant intensification of zonal wind between Iceland and Scandinavia, though the area of significance is limited (Fig. 13k). These changes can be associated with zonal wind changes that mainly occur during winter and to a lesser extent during summer (Figs. S15j-1 and S16j-1). The zonal wind changes over the northern Atlantic are most likely driven by SST changes and to a lesser extent by sea ice loss (Smirnov et al., 2015; Köhler et al., 2024; Wills et al., 2024), although the mechanisms for the differences between ST1 and ST2 as well as the relative contributions of SST and sea ice changes are unknown and therefore require more research in the future. Finally, the enhanced westerly moisture transport in ST1 and ST2 is accompanied by increased moisture convergence along the Scandinavian mountain ranges and is indicative of an intensification of precipitation over the respective region (Fig. 13b-c, h-i). However, the differences between POLARRES and NE30 are largely insignificant, which could be partly due to the relatively short time period (10 years) over which CESM model data are available. The short time period reduces the power of statistical analyses (i.e. due to the lower degrees of freedom), which may partly explain the absence of statistical significance. Therefore, the enhanced westerly moisture transport and convergence can only partially explain why the differences in precipitation between NE30 and POLARRES are more pronounced over Scandinavia and less so in other regions. It is likely that other drivers (e.g. orographic-induced updrafts and convection) also influence the differences in precipitation between NE30 and POLARRES. However, to understand how the drivers of precipitation change with grid refinement under different SST and sea ice regimes requires further research.

#### 4 Discussion and Conclusions

We have evaluated and assessed present-day and future temperature and precipitation extremes over the Arctic, using the variable-resolution Community Earth System Model version 2.2 (VR-CESM). We applied a globally uniform 1° CESM grid (NE30) and a dual-polar VR grid with regional grid refinements to 28 km over the Arctic and Antarctica (POLARRES). Using both model grids, we run three different types of simulations with interactively coupled atmosphere and land surface models, and prescribed sea ice and sea surface temperatures, namely: a 30-year simulation (1985–2014) used to evaluate the model grids, and 10-year present-day (2005–2014) and future (2090–2099) simulations used to assess future changes in temperature and precipitation extremes. Here, the 10-year present-day and future simulations follow two storylines of Arctic summer climate change representing a combination of strong/weak Arctic atmospheric warming at the 850 hPa level and weak/strong sea surface warming in the Barents-Kara Seas. To evaluate and assess present-day and future temperature and precipitation extremes, we analyzed eight extreme metrics, including four extreme metrics each for temperature and precipitation.

The evaluations show that both NE30 and POLARRES have climatological higher temperatures over Europe and parts of Greenland, and climatological lower temperatures over the central Arctic, northeastern Canada, and parts of Greenland, with lower temperatures in POLARRES than in NE30. For temperature extreme metrics, POLARRES performs slightly better in simulating annual maximum temperature, while NE30 performs better for other temperature extreme metrics (annual mimimum temperature and warm and cold spell duration). Spatially, the cold and warm temperature differences in annual minimum and maximum temperature correspond well to the spatial patterns of the climatological biases, but the magnitude of the extreme temperature differences is much larger. The lower temperatures can be partly explained by large warm near-surface temperature biases in the reanalysis products, caused by poor representation of snow over sea ice and the thickness of the sea ice itself. Additionally, the lower temperatures may be related to overestimated cloud cover (relative to the reference data) leading to excessive reflection and cooling, and increased summer surface albedo and snow depth over northeastern Canada and the central Arctic in POLARRES. However, to understand the magnitude of the lower temperatures in CESM, validation of near-surface temperature against meteorological observations in the Arctic is needed and thus recommended for future research.

NE30 and POLARRES have climatological dry precipitation differences over most of continental mid-latitudes and climatological wet precipitation differences over the oceans and most of the Arctic land regions. The dry precipitation differences indicate a precipitation deficit over land, which, to some extent, may contribute to the higher temperatures over the land surface via a soil moisture deficit and associated reduced evaporative cooling. POLARRES performs slightly better than NE30 in simulating the precipitation climatology, with higher precipitation rates over mountainous terrain in Scandinavia, southeastern Greenland, and Alaska, which is mainly due to the higher grid resolution and better resolved topography. With respect to precipitation extremes, the performance of POLARRES and NE30 depends on the performance criteria (i.e. RMSD or AVGD) and the data used as reference (ERA5, JRA-3Q or RACMO) for the evaluation. Using the RSMD as the performance criterion, POLARRES generally performs better than NE30 in simulating the 99<sup>th</sup> percentile of daily precipitation and the number of heavy precipitation and consecutive dry days, while NE30 performs better in simulating highest 5-day precipitation sums. The larger highest 5-day precipitation sum differences in POLARRES may partly be related to a higher frequency and intensity of

storm systems bringing significant amounts of precipitation over a period of several days. However, more research is needed to understand the impact of grid refinement on the frequency and intensity of storm systems in the Arctic as well as its effect on precipitation extremes in CESM.

Future projections suggest that the annual mean, maximum and minimum temperature, and the warm spell duration will mostly increase in magnitude or duration in all regions of the Arctic, while the cold spell duration is projected to decrease. Here, the rate of change varies by region and the applied storyline. The projected increases are larger for ST2, with the largest increases being projected over the currently sea ice-dominated central Arctic. The larger increases in ST2 suggest that sea surface warming and associated sea ice loss over the Arctic have a stronger response than Arctic low-level tropospheric warming, especially over the sea ice itself. In ST1, increases in temperature extremes are especially smaller over the ocean and Scandinavia, and are accompanied by a local decrease in annual mean temperature due to projected cooling anomalies over the North Atlantic. Regarding the differences in projected changes by NE30 and POLARRES, the differences are more pronounced for annual maximum temperature with larger increases in POLARRES over northeastern Canada, which is mainly due to reduced snow depth. The annual maximum temperature differences over other land regions, such as the larger increases in Scandinavia or the smaller increases in Siberia, could be related to either stronger or weaker warming over the Eurasian continent.

Precipitation means and extremes are mostly projected to increase in intensity and frequency, with the projected changes in extremes being greater than the projected changes in mean precipitation. Only over the northern Atlantic, annual mean precipitation is projected to decrease due to local cooling-induced decreases in atmospheric moisture and precipitation. The projected changes are mostly stronger for ST2, which can be associated with the stronger warming-induced increases in atmospheric moisture and precipitation. The projected changes in consecutive dry days suggest that relatively dry regions with a high present-day number of consecutive dry days (central Arctic, north(east)ern Greenland and Canada, and eastern Siberia) will become wetter in a warming climate, resulting in a decreased number of consecutive dry days. Relatively wet regions with a low present-day number of consecutive dry days (e.g., Scandinavia) are expected to become wetter in a warming climate but may also experience longer periods of droughts in summer, resulting in an increased number of consecutive dry days.

Regarding the differences in regional projected changes by NE30 and POLARRES, the differences often show mixed signals depending on the extreme metrics and region but are especially more pronounced over Scandinavia as well as for the highest 5-day precipitation sums. Here, POLARRES projects larger increases in highest 5-day precipitation sums, which may be partly due to the higher frequency and intensity of storm systems associated with higher resolution grids. Further, POLARRES projects larger increases in the 99<sup>th</sup> percentile of daily precipitation, the highest 5-day precipitation sums, and heavy precipitation days over Scandinavia, which may be partly due to enhanced westerly moisture transport and convergence over the respective region, and possibly due to other drivers, such as orographic-induced updrafts and convection.

This study shows that, relative to the reference data used, VR-CESM generally outperforms the coarse-resolution CESM grid in simulating precipitation climatology and extremes. In contrast, the coarse-resolution CESM grid often performs better in simulating temperature climatology and extremes. This suggests that VR-CESM can be a valuable tool for simulating present-day and future extremes in the Arctic. However, model improvements are needed to, for instance, reduce temperature

differences relative to reference data. It is important to note that the evaluation of model outcomes primarily relied on reanalysis

fields, which can be strongly influenced by the underlying forecast model in regions with little or no observations. Therefore,

incorporating meteorological observations and remote sensing data may help to better understand the differences between

CESM model output and observational references. Additionally, further research is needed to better understand the physical

drivers and mechanisms of changes in Arctic temperature and precipitation means and extremes, whether induced by warming

or by the use of grid refinement.

Finally, this study demonstrates that the projected changes in Arctic temperature and precipitation extremes can vary sub-

stantially depending on the storyline applied. In this study, the average difference in near-surface temperature projections

between ST1 and ST2 is about 2.5-3 °C by the end of the  $21^{st}$  century. However, this is considerably narrower than the  $5^{th}$ -

95th percentile range of approximately 9 °C in Arctic surface temperature change projected by CMIP6 models under SSP5-8.5

(Lee et al., 2021). The narrower range of projected outcomes suggests that the storyline approach used here may support a

more targeted understanding of future Arctic temperature and precipitation extremes, and could help inform adaptation and

mitigation strategies in response to climate extremes in the region.

Data availability. Publicly available data will be stored in a data archive on Zenodo (https://doi.org/10.5281/zenodo.14961708) upon pub-

lication. The data archive contains the daily near-surface temperature and precipitation output and calculated temperature and precipitation

extreme metrics.

610

620

Author contributions. RRW and WJB designed the study. RRW ran the model with technical support of ARH. XJL developed the Arctic

storylines. CTD provided the RACMO data used for evaluation of the model output. RRW analyzed the model results and prepared the text

and figures. WJB supervised RRW. All authors contributed to the final text.

Competing interests. The contact author has declared that none of the authors has any competing interests.

Disclaimer. TEXT

Acknowledgements. We acknowledge the support of PolarRES (grant no. 101003590), a project of the European Union's Horizon 2020

research and innovation programme. Adam R. Herrington has been supported by the NSF National Center for Atmospheric Research, which

is a major facility sponsored by the National Science Foundation (grant no. 1852977). Computing and data storage resources, including the

Chevenne supercomputer (https://doi.org/10.5065/D6RX99HX, Computational and Information Systems Laboratory, 2019), were provided

by the Computational and Information Systems Laboratory (CISL) at NSF NCAR.

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

**Figure 1.** Schematic representation of the variable resolution (VR) spectral element grid (**a**) with regional grid refinements over the Arctic (**b**), and an overview of regions used for analysis throughout this study (**c**), which include four land regions AWC (Alaska and Western Canada; yellow), ECG (Eastern Canada and Greenland; orange), SCA (Scandinavia; red), and SIB (Siberia; green), a sea ice region (SIC; blue), and an ocean region (OCN; cyan).

**Figure 2.** Present-day B (2005–2014) state (**a-b,e-f**) and future (2090–2099) changes (**c-d,g-h**) in northern hemisphere (poleward of 50°N) sea surface temperature (SST (°C); **a-d**) and sea ice cover (SIC (%); **e-h**), as output by the NE30 grid for two different storylines: ST1 (**a,c,e,g**) and ST2 (**b,d,f,h**). The future changes are expressed as absolute differences relative to present-day.

**Figure 3.** Present-day A (1985–2014) northern hemisphere (poleward of 50°N) annual mean 2m temperature (°C) differences between NE30 and ERA5 (**a**), JRA-3Q (**c**) or RACMO (**e**), and between POLARRES and ERA5 (**b**), JRA-3Q (**d**), or RACMO (**f**). The area-weighted root-mean-square-difference (RMSD) and average difference (AVGD) listed above the panels are calculated for the domain 60-90°N. The stippling represents the significance of temperature differences at the 95% confidence level.

**Figure 4.** Present-day A (1985–2014) daily 2m temperature (°C) probability density functions (PDFs) for NE30 (red), POLARRES (blue), ERA5 (black), JRA-3Q (green), and RACMO (orange). The PDFs are calculated on a NE30 grid for the six different regions as shown in Fig. 1.

Figure 5. Present-day A (1985–2014) northern hemisphere (poleward of  $50^{\circ}$ N) annual maximum temperature (TXx (°C); **a-d**), warm spell duration index (WSDI (d yr<sup>-1</sup>); **e-h**), annual minimum temperature (TNn (°C); **j-l**), and cold spell duration index (CSDI (d yr<sup>-1</sup>); **m-p**) state (1<sup>st</sup> column) and differences between NE30 and ERA5 (2<sup>nd</sup> column), POLARRES and ERA5 (3<sup>rd</sup> column), and POLARRES and NE30 (4<sup>th</sup> column). The area-weighted root-mean-square-difference (RMSD) and average difference (AVGD) listed above the 2<sup>nd</sup>-4<sup>th</sup> column panels are calculated for the domain 60-90°N. The stippling represents the significance of differences at the 95% confidence level.

**Figure 6.** Same as Fig. 3, but for annual mean precipitation  $(mm d^{-1})$ 

**Figure 7.** Same as Fig. 4, but for daily precipitation rates  $(mm d^{-1})$ 

**Figure 8.** Same as Fig.5, but for the  $99^{th}$  percentile of daily precipitation (P99 (mmd<sup>-1</sup>); **a-d**), highest 5-day precipitation (RX5day (mm); **e-h**), heavy precipitation days (R10mm (dyr<sup>-1</sup>); **i-l**), and consecutive dry days (CDD (dyr<sup>-1</sup>); **m-p**)

**Figure 9.** Northern hemisphere (poleward of  $50^{\circ}$ N) present-day B (2005-2014) annual 2m temperature and projected future (2090-2099) changes in annual 2m temperature ( $^{\circ}$ C) for NE30 (**a-d**) and POLARRES (**e-h**) in ST1 ( $1^{st}$  and  $2^{nd}$  columns) and ST2 ( $3^{rd}$  and  $4^{th}$  columns). The future changes are expressed as absolute differences relative to present-day. The differences between NE30 and POLARRES are shown in the bottom row (**i-l**). The stippling represents the significance of differences at the 95% confidence level.

Figure 10. Projected future (2090–2099) changes in annual maximum temperature (TXx ( $^{\circ}$ C); **a**), annual minimum temperature (TNn ( $^{\circ}$ C; **b**), warm spell duration index (WSDI ( $^{\circ}$ dyr<sup>-1</sup>); **c**), and cold spell duration index (CSDI ( $^{\circ}$ dyr<sup>-1</sup>); **d**) for six different regions as shown in Fig. 1. The bars show the projected changes in NE30 ( $^{st}$  and  $^{rd}$  bars) and POLARRES ( $^{st}$  and  $^{the}$  bars) for ST1 ( $^{tt}$  and  $^{tt}$  bars) and ST2 ( $^{tt}$  and  $^{tt}$  bars). The black dots denote the projected changes in annual mean near-surface temperature ( $^{\circ}$ C). The error bars (black) represent the 95% confidence interval.

Figure 11. Same as Fig. 9, but for annual mean precipitation  $(\mathrm{mm}\,\mathrm{d}^{-1})$ 

**Figure 12.** Same as Fig. 10, but for the  $99^{th}$  percentile of daily precipitation (P99 (mm d<sup>-1</sup>); **a**), highest 5-day precipitation (RX5day (mm); **b**), heavy precipitation days (R10mm (dyr<sup>-1</sup>); **c**), and consecutive dry days (CDD (dyr<sup>-1</sup>); **d**)

Figure 13. Projected future (2090–2099) changes in northern hemisphere (poleward of  $30^{\circ}$ N) vertically-integrated moisture flux convergence (VIMFC (g/m2 – s); shading) and 850 hPa moisture flux ((m/s)/ (g/kg); vectors) (a-c,g-i), and 850 hPa zonal wind (d-f,j-l; m/s; shading), as output by NE30 ( $1^{st}$  column) and POLARRES ( $2^{nd}$  column) for ST1 (a-f) and ST2 (g-l). The differences between NE30 and POLARRES are shown in the  $3^{rd}$  column. The zonal wind contours denote the climatological mean of the present-day zonal wind at 850 hPa, where red stands for positive (eastward) and blue stands for negative (westward). The stippling and the vectors (black) represent the significance of future changes at the 95% confidence level. The grey vectors represent the changes in moisture flux that are insignificant.

**Table 1.** Metrics of daily temperature and precipitation extremes used in this study

| Extreme Metric                                     | Index                               | Description                                                                                                                                                                                                                                                                                                                                                                                                                                                            |
|----------------------------------------------------|-------------------------------------|------------------------------------------------------------------------------------------------------------------------------------------------------------------------------------------------------------------------------------------------------------------------------------------------------------------------------------------------------------------------------------------------------------------------------------------------------------------------|
| Temperature extremes                               |                                     |                                                                                                                                                                                                                                                                                                                                                                                                                                                                        |
| Annual maximum temperature                         | TXx (°C)                            | Annual maximum of daily maximum temperature (TX)                                                                                                                                                                                                                                                                                                                                                                                                                       |
| Annual minimum temperature                         | TNx (°C)                            | Annual minimum of daily minimum temperature (TN)                                                                                                                                                                                                                                                                                                                                                                                                                       |
| Warm spell duration                                | WSDI $(\mathrm{d}\mathrm{yr}^{-1})$ | Warm spell duration index, defined as the annual number of days in                                                                                                                                                                                                                                                                                                                                                                                                     |
| Cold spell duration                                | CSDI $(\mathrm{d}\mathrm{yr}^{-1})$ | intervals of at least 6 consecutive days where TX > TX90, where TX90 is the $90^{th}$ percentile of daily maximum temperature, calculated for each calendar day using a running window of 15 days Cold spell duration index, defined as the annual number of days in intervals of at least 6 consecutive days where TN < TN10, where TN10 is the $10^{th}$ percentile of daily minimum temperature, calculated for each calendar day using a running window of 15 days |
| Precipitation extremes                             |                                     |                                                                                                                                                                                                                                                                                                                                                                                                                                                                        |
| 99 <sup>th</sup> percentile of daily precipitation | P99 $(\text{mm d}^{-1})$            | 99 <sup>th</sup> percentile of daily precipitation sums                                                                                                                                                                                                                                                                                                                                                                                                                |
| Highest 5-day precipitation sum                    | RX5day (mm)                         | Highest amount of precipitation over an interval of 5 days                                                                                                                                                                                                                                                                                                                                                                                                             |
| Number of heavy precipitation days                 | $R10mm~(\mathrm{dyr}^{-1})$         | Number of days with heavy precipitation, defined as daily precipitation that is equal to or higher than $10\mathrm{mm}\mathrm{d}^{-1}$                                                                                                                                                                                                                                                                                                                                 |
| Number of consecutive dry days                     | $CDD (dyr^{-1})$                    | Greatest number of consecutive days with daily precipitation less than $1\mathrm{mm}\mathrm{d}^{-1}$                                                                                                                                                                                                                                                                                                                                                                   |