# Peer review of "Arctic temperature and precipitation extremes in present-day and future storyline-based variable resolution Community Earth System Model simulations"

_EGUsphere, 2025_

## Referee Comment (RC2)

**General comments**

This paper addresses temperature and precipitation extremes in the Arctic in CESM experiments with prescibed SST and sea ice. Two versions of CESM, one with uniform resolution and another with a refined grid at high latitudes are considered. First, comparisons with reanalysis and regional model datasets are presented for near-present climate simulations, and second, future changes in extremes are evaluated for two storylines in a high-emission scenario.

The results show that in warmer climate conditions, high temperature extremes and precipitation extremes will become stronger and/or more frequent, while cold/dry extremes will generally become weaker and/or less frequent in the Arctic region. Aside from these somewhat expected findigs, there are some differences between the storylines and some (mainly subtler) differences between the experiments performed with the two different grids.

This paper represents an interesting application of the storyline approach to the future Arctic climate, also documenting how the results depend on the model grid. Overall, the paper is thorough and well-written. The technical quality of figures is good, but statistical testing is mostly missing.

**Major comments**

1. The evaluation of statistical significance of the findings is largely lacking in this work. In most of the figures, it is not considered at all, and while it is included in some (e.g., Figs. 3 and 13), the method for calculating the statistical significance is not explained. For example, does the statistical testing account for the multiplicity problem? That is, when tests are performed for a large number of grid points, some of them would likely show nominally significant differences even if the results were generated randomly. See, for example:

Wilks, D. S., 2016: "The Stippling Shows Statistically Significant Grid Points": How Research Results are Routinely Overstated and Overinterpreted, and What to Do about It. Bull. Amer. Meteor. Soc., 97, 2263–2273, https://doi.org/10.1175/BAMS-D-15-00267.1.

Testing the statistical significance is especially relevant because the climate change signal is defined based on relatively short runs (10 years) and because extremes are more subject to the effects of internal variability than mean values.

While, for example, the general increase of warm extremes and wet extremes at most Arctic locations might be robust, it is less obvious how robust are the differences between the changes in extremes for the two storylines, let alone the corresponding differences between POLARRES and and NE30 runs. Indeed, it is obvious that some of the figures contain mostly noise (Fig. S13f,h being a particularly "good" example).

I recommend that the statistical significance of the results should be evaluated more systematically, and the reporting should be mainly focused on those aspects found robust. In the case of areal-mean results, such as those Fig. 10 and 12, uncertainty could be illustrated by showing confidence intervals. Just be careful with their interpretation:

Lanzante, J. R., 2005: A Cautionary Note on the Use of Error Bars. J. Climate, 18, 3699–3703, https://doi.org/10.1175/JCLI3499.1.

2. The manuscript is not fully clear about how regridding of the results from different model runs and reanalysis to a common grid is made. For example, is it made using conservative interpolation (which conserves area means) or bilinear interpolation between the nearest grid points, or some other approach? Also, it is not stated clearly, whether regridding is applied to the original temperature and precipitation values before calculating the extremes, or whether the extremes are first calculated at the original model resolution and then regridded to the common grid for plotting.

These choices are especially important for precipitation extremes. In particular, I would expect that if (1) bilinear (rather than conservative) interpolation is used and/or (2) the extremes are calculated before regridding, there would be a systematic increase in high precipitation rates (e.g., P99) with improving resolution, simply because daily precipitation values show substantial small-scale variations. On the other hand, if extremes are calculated after a conservative regridding is applied, it is not obvious this will happen (or at least, there would be no trivial reason for that).

There is not necessarily a single "right" solution to how this matter should be handled, but at any rate, you should be clear about how you do it, and justify your choice. See also minor comments 11 and 12.

**Minor comments**

1. lines 13 and 102: replace the latter "strong/weak" with "weak/strong"? The idea of ST1 and ST2 is to contrast a case with strong land warming and weak SST warming with a case with weak land warming and strong SST warming.

2. line 16: It could be mentioned that the better performance of the $1°$ grid in simulating temperature extremes is related to a larger negative temperature bias in the VR grid. (Incidentally, I think this might be a matter of model tuning rather than any fundamental issue associated with higher resolution).

3. line 124: You can delete "cloud-aerosol" before "radiation scheme".

4. Some parts of the model and experimental description appear unnecessarily detailed in the context of this work. In particular, the description of CLM5 on lines 132–143 and land surface treatment on lines 216-223.

5. lines 176–177: at least in CICE4, increasing the parameter $r_{snw}$ actually decreases the snow grain radius over sea ice, and therefore, increases snow albedo. Please check this.

6. lines 182–184: Are microphysical substep and microphysical timestep the same thing or not?

7. lines 198–203: Please use a notation that it consistent with Levine et al. (2024). Storyline ST1 corresponds to storyline D (not B2) in Levine et al., denoted as ArcAmp+/BKWarm- (not PolAmpl+BKSSTWarm-). Similarly, ST2 corresponds to storyline A (not B1) denoted as ArcAmp-/BKWarm+ in Levine et al.

8. line 213: It would be useful to mention the global-mean near-surface temperature change from the present-day (2005–2014) to the future period (2090–2099) for ST1 and ST2.

9. Sect. 2.3. Mention that in regions with sparse observations, such as large parts of the Arctic, the reanalysis fields are strongly influenced by the underlying forecast model.

10. line 307–309: If/when CSDI is calculated based on CESM's own climatology, there is no reason why a cold bias would lead to a positive CSDI bias. So the negative CSDI bias is note particularly conunter-intuitive.

11. lines 348–357: If the precipitation PDFs are based on daily precipitation rates

regridded to the NE30 grid (as stated), then what explains the systematic change in the PDFs (more frequent very high values) with improving horizontal resolution?

12. As a follow-up comment, if Fig. 7 is based on (e.g.) bilinearly interpolated precipitation data, it would be interesting to see how the PDFs behave if conservative interpolation is used instead.

13. lines 368–369 (also 552–553). Please check if the cited references support your statement regarding extra-tropical storm systems. Based on a cursory reading, they seem to focus on low-latitude systems.

14. lines 458–461: It is not clear what these "average" absolute and relative increases represent. The ranges quoted do not cover the actual range of values in Fig. 12. A simple approach would be to take averages over 60-90°N.

15. line 547–549: A possible physical interpretation of this is that the regions with high present-day CDD are very high-latitude regions with cold winters, in which daily precipitation above 1 mm is relatively rare (but will increase with warming), while the regions with lower present-day CDD are in the southern parts of Arctic and might experience longer dry periods in summer in a warmer climate.

16. line 566: "this narrows the uncertainty range". This requires a more careful reasoning. The definition of storylines in Levine et al. (2024) is based on Arctic land and Barents-Kara Sea warming *normalized* by global-mean warming. Therefore, while the storyline approach is invaluable for impact studies, it is not obvious to me how it reduces the uncertainty related to the overall Arctic warming.

17. Figures 3, 5, 6, 8, S1, S2, S3, S5 and S6: Given that even the renalysis are affected strongly by the underlying model, I suggest to replace root-mean-square error (RMSE) with root-mean-square differene (RMSD), and "Bias" with average difference (AVG or AVGD).

18. Figs. 3 and 6: I suggest to add one more panel showing the difference POLARRES−NE30.

19. Fig. 4: Are these PDFs based on daily-mean data? Please mention that in the caption.

20. Figs. 13 and S15: "... vectors represent the significance". Does this mean that vectors are drawn only where the change is significant (in the third column, they seem to be drawn everywhere!)?

21. Fig. S4: Explain the meaning of stippling.

**Language and technical corrections**

1. line 307. Replace "decrease in the number" with "smaller number".

2. line 318: Replace "(TXx)" with "and TXx".

3. lines 352–353: "A possible explanation for the lower extreme precipitation ... is that extreme precipitation rates are underestimated". This could be shortened: "It is possible that extreme precipitation rates are underestimated in ERA5 and JRA-3Q".

4. line 380: "although CESM tends to be slightly drier". You presumably mean "although the dry bias for CESM appears slightly larger"? The CESM results remain unaltered, only the reference changes ...

5. lines 414–415 and 430–432. The use of parentheses to shorten sentences often makes the text more difficult to read. It should be avoided especially in cases in which parentheses are also used in their conventional purpose. See

*https://eos.org/opinions/parentheses-are-are-not-for-references-and-clarification-saving-space*.

6. line 419: "response on temperature" should be "effect on temperature"?

7. line 472: replace "wetter and experience" with "wetter but experience"?

8. line 496: "Fig. 13b-c,e-f". Double-check that these are the correct figure and panel numbers.

9. line 519: replace "increased cloud cover" with "overestimated cloud cover".

10. caption of Fig. 1: SIC and OCN are marked with colours rather than diagonal and crossed pattern.

11. Fig. S7: "Same as Figs. S8 and S9". It is not reader-friendly to refer to later figures in the caption. Move Fig. S7 after S8 and S9?

---

## Author Response (AR1)

We gratefully acknowledge the reviewer for his/her remarks and suggestions, which improved the quality of the manuscript significantly. We have carefully considered the suggestions of the reviewer and we provide a point-by-point response to the reviewer's comments. For clarity, the reviewer's comments are given in bold italics and the responses are given in plain text. References that do not refer to those in the main manuscript are listed below. The manuscript will be modified according to the responses that are given to the comments.

This article examines changes to mean and extreme metrics of temperature and precipitation in VR CESM simulations, where mesh refinement is over the poles. Comparisons are made with standard resolution CESM output. Simulations of present day and of two storylines of future change based on other work are performed with and without mesh refinement over the region of interest. They find that high extremes of temperature and precipitation will increase in many of the study regions encompassing high latitude land and ocean.

This is a well written and thorough paper. I think that the application of different storylines of future change within a high-resolution model is a novel and interesting way to use high-resolution modelling at a lower computational cost. Throughout there is thought put into trying to physically understand the results, and with trying to square these results with past studies, placing it nicely within the larger body of literature. Figures are high quality and contain a lot of information that is well presented.

I have mainly minor comments I'd like to see addressed before publication.

We thank the reviewer for his/her evaluation of the manuscript. We have tried to address all concerns.

**General:**

In terms of readability, using capital letters to refer to simulation names (e.g. POLARRES), regions (e.g ECG, SCA..), and metrics of extremes (e.g. WSDI, CSDI) did at times make it a bit difficult to understand. Using bolding, italics, or perhaps single quotation marks to differentiate between them (depending on what the journal might allow), or even simply spelling out the whole name of, for example, a region, would improve ease of reading.

We will improve the readability of the manuscript by spelling out the abbreviations that are related to the temperature and precipitation extreme metrics and the regions as these are used most throughout the manuscript. For example, WSDI will then become warm spell duration and SIC will become the sea ice region.

The Data & Methods are very comprehensive but probably a bit more specific and detailed than is necessary for this paper, more suited toward a paper published in a journal about geophysical model development. For example, details of CLM5, computational costs, parameter tuning, where reanalysis data is retrieved from, an unused dataset, etc could be pared down to the necessary information not available in other published work.

We will shorten the *Data and Methods* section by shortening the subsections related to the model description (especially the model description of CLM5) as most of the details described in these subsections can also be found in other published work. As for the other parts of the Data and Methods section we will have a thorough look onto which parts of the main text can be shortened.

I think it's important to keep in mind that agreement/disagreement between a reanalysis product and a model over regions of sparse observations should always be taken with a grain of salt, and that reanalyses are not equivalent to observations. Treatment of differences between a model and a reanalysis product in some regions is akin to comparing model and model, and reanalyses themselves have many biases. Perhaps a bit more discussion to this end would be beneficial.

We agree with the reviewer that the differences between the CESM model output and reanalysis data need to be interpreted with care as reanalysis data can also contain many biases, especially in regions with sparse observations where reanalysis fields can be strongly influenced by the underlying forecast model. For example, in L319-328 of the original manuscript, we already indicate that the cold temperature differences in CESM is partly related to large warm near-surface temperature biases that have been found over sea ice during winter (clear sky) conditions in reanalysis datasets, such as ERA5 and JRA55 (the predecessor of JRA-3Q). These warm biases are mainly caused by a poor representation of snow on top of the sea ice and the sea ice thickness itself and can exaggerate the cold temperature differences in CESM. To understand the magnitude of cold temperature differences, validation of near-surface temperature against meteorological observations are therefore recommended in future research.

As the reanalysis data can be strongly influenced by the underlying model in regions with sparse observations, we have chosen to describe model-related differences, such as those between CESM and ERA5, in terms of differences rather than errors and biases. We will therefore replace "root-mean-square-error (RMSE)" and "Bias" with "root-mean-square-difference (RMSD)" and "average difference (AVGD)", respectively, and we will adjust the wording in the main text following this terminology (e.g. cold temperature bias becomes cold temperature difference). Additionally, we will add an extra sentence to Section 2.3 *Model evaluation* to mention that the reanalysis fields can be strongly influenced by the underlying model in regions with sparse observations. We will also add an extra sentence to the *Discussion and Conclusions* section to emphasize that comparisons with meteorological observations are beneficial for a better understanding of the temperature and precipitation differences in CESM model output.

**Specific/Technical**

L 13 & elsewhere: 'SST warming' should either say 'sea surface warming' or 'SST increase' because a temperature can't warm.

We will replace "SST warming" with "sea surface warming" throughout the manuscript.

L30-32: I don't think this is a complete treatment of the literature on whether cold extremes are becoming more common in the mid-latitudes, as more recent studies (e.g. Cohen et al 2023, Van Olgdenberg 2019, Blackport 2024) have not come to the conclusion that cold extremes are occurring more often. Additionally, it's not particularly relevant to the regions discussed later in the paper so probably best to just not include it.

We agree with the reviewer that the treatment of the literature in this sentence is not complete since recent studies (e.g., van Oldenborgh et al., 2019, Cohen et al. 2023, Blackport & Fyfe, 2024) as mentioned by the reviewer indeed indicate that there is no clear trend (Cohen et al., 2023) or decreasing trend (van Oldenborgh et al., 2019; Blackport & Fyfe, 2024) in the occurrence of winter cold extremes in the mid-latitudes. As our study mainly focusses on the Arctic, we therefore will follow the reviewer's suggestion by not including this sentence in the manuscript.

Paragraph beginning at line 81: Might want to add references to Morris et al papers from 2023-2025 using VR CESM to look at wind extremes. They find that sometimes the low resolution CESM gets a different sign of response than VR CESM for extreme winds, and perhaps it's something worth looking into for the Arctic region as well.

We will include the references to the work of Morris et al. (2023, 2024) and Morris and Kushner (2025) in the manuscript. Regarding wind extremes in the Arctic region, we believe that investigating the impact of regional grid refinement on the projection of future wind extremes would be interesting as well. However, as the manuscript is already quite lengthy, this could be a topic for future studies.

**L95: I'm curious why there is an enhancement of resolution over the Antarctic as well?**

The POLARRES grid is originally designed for Work Package 2 of the PolarRES project (<a href="https://polarres.eu">https://polarres.eu</a>). The project itself studies the interactions between the atmosphere, oceans, and sea ice in the Arctic and Antarctica using global climate models and regional climate models, amongst others. In Work Package 2 of the PolarRES project, three global variable-resolution climate models (CESM, ICON, and MPAS) are used to improve our understanding of atmospheric teleconnections in both the northern and southern hemispheres, regional climate impacts over the Arctic and Antarctic, and how polar climate systems affect and are affected by lower latitudes. The CESM simulations performed for this study also contribute to the research conducted in this work package. As the focus is also on Antarctica in this work package, there is an enhancement of resolution over the Antarctic as well.

**L283-285: difficult sentence to understand, I'd suggest re-writing to clarify.**

We will rewrite the sentence to "The cold temperature differences are largest compared to ERA5, with average differences of -1.32 °C for NE30 and -1.72 °C for POLARRES (Figs. 3a-b). The differences are smaller compared to RACMO, with average differences of -1.24 °C for NE30 and -1.64 °C for POLARRES (Figs. 3e-f), and compared to JRA-3Q, with average differences of -0.54 °C and -0.95 °C, respectively (Figs. 3c-d)".

**L353: missing comma before 'as for example'**

We will add a comma in the main text before "as for example".

**L408: I think here and below should read 'cold' anomalies and not 'cooling' anomalies.**

We agree. We will replace "cooling" anomalies with "cold" anomalies.

**L471: I find this sentence a bit confusing. Do wet regions become more variable?**

The sentence "This means that relatively dry regions will become wetter, while relatively wet regions will become wetter and experience longer periods of drought." can be explained as follows. Relatively dry regions are very high-latitude regions with cold winters, in which daily precipitation above 1 mm is relatively rare. This can explain why these regions have a high present-day number of consecutive dry days. With increased warming, these regions experience an increase in precipitation and a decrease in consecutive dry days, suggesting that these regions will become wetter. Relatively wet regions (i.e. the regions with a lower present-day number of consecutive dry days) are found in the warmer southern parts of the Arctic (i.e. at lower latitudes) and experience more precipitation. These regions might experience longer periods of drought in summer in a warmer climate, resulting in an increase in the number of consecutive dry days. However, at the same time, these regions will also become wetter, as the mean precipitation and precipitation extremes will increase in intensity and frequency. This results in a higher precipitation variability, particularly over regions, such as Scandinavia and over lower latitudes. We will rephrase the sentences in the manuscript to clarify our findings.

**L500: comma after regimes, remove 'as well' from end.**

• We will add a comma after "regimes" and remove "as well".

**L532: Does POLARRES produce more storms/more strong storms than the reanalyses do for this region?**

Although we did not examine the frequency and intensity of storm systems in POLARRES, a research project is currently investigating the impact of regional grid refinement on the Northern Hemisphere extra-tropical storm tracks in three global models, which are CESM, ICON and MPAS (personal communication from Kajsa Parding at the Norwegian Meteorological Institute). The initial findings of this project indicate that, compared to ERA5, the NE30 grid underestimates the number of extra-tropical storms. The POLARRES grid generates more extra-tropical storms than the NE30 grid, particularly over the North Atlantic region. On average, the POLARRES grid also generates more extra-tropical storms than ERA5 in summer, with positive differences observed over Greenland, the North American mid-latitudes, and (north)eastern Asia. The latter two regions overlap with those where larger differences in the highest 5-day precipitation sum are observed. However, these regions also show more storm tracks than ERA5 in NE30. This suggests that the higher frequency and intensity of storm systems with increasing resolution cannot fully explain the larger highest 5-day precipitation sum differences in POLARRES. Therefore, we will adjust the text in lines 368-369 of the original manuscript as follows:

"The larger differences in POLARRES may be partly due to a higher frequency and intensity of storm systems, which bring significant amounts of precipitation over a period of several days (Zarzycki and Jablonowski, 2014).",

and in lines 531-532 as follows:

"The larger highest 5-day precipitation sum differences in POLARRES may partly be related to a higher frequency and intensity of storm systems bringing significant amounts of precipitation over a period of several days. However, more research is needed to understand the impact of grid refinement on the frequency and intensity of storm systems in the Arctic as well as its effect on precipitation extremes in CESM."

**L547-549: Is this sentence at odds with that at L471?**

The sentence here is not at odds with that at L471. What we suggest here is that regions with a high number of present-day consecutive dry days show a decrease in the number of consecutive dry days. These regions are relatively dry with cold winters in which daily precipitation above 1 mm is relatively rare. Under a warmer climate, precipitation will increase, resulting a in lower number of consecutive dry days. This means relatively dry regions will become wetter. Regions with a low present-day number of consecutive dry days – located in the southern part of the Arctic – can be considered as relatively wet and show an increase in the number of consecutive dry days, which is presumably related to prolonged periods of drought during summer. However, these regions will also become wetter under a warming climate, which is indicated by the increase in intensity and frequency of precipitation means and extremes. We will rephrase this sentence in the manuscript to clarify our findings.

**Figure 2 caption & elsewhere: 'outputted' -> 'output'**

We will replace "outputted" with "output" here and elsewhere in the manuscript.

**References**

Blackport, R., Fyfe, J.C. Amplified warming of North American cold extremes linked to human-induced changes in temperature variability. *Nat Commun* **15**, 5864 (2024). <a href="https://doi.org/10.1038/s41467-024-49734-8">https://doi.org/10.1038/s41467-024-49734-8</a>.

Cohen, J., Agel, L., Barlow, M. *et al.* No detectable trend in mid-latitude cold extremes during the recent period of Arctic amplification. *Commun Earth Environ* **4**, 341 (2023). https://doi.org/10.1038/s43247-023-01008-9.

Morris, M., Kushner, P.J., Moore, G.W.K. *et al.* Atmospheric circulation patterns associated with extreme wind events in Canadian cities. *Journal of Climate*, **36(13)**, 4443-4460 (2023). https://doi.org/10.1175/JCLI-D-22-0719.1.

Morris, M., Kushner, P.J., Moore, G.W.K. *et al.* Resolution Dependence of Extreme Wind Speed Projections in the Great Lakes Region. *Journal of Climate*, **37(11)**, 3153-3171 (2024). <a href="https://doi.org/10.1175/JCLI-D-23-0547.1">https://doi.org/10.1175/JCLI-D-23-0547.1</a>.

Morris, M., & Kushner, P.J. Mechanisms of the Extreme Wind Speed Response to Climate Change in Variable-Resolution Climate Simulations of Western, Central, and Atlantic Canada. *Journal of Climate*. (2025). <a href="https://doi.org/10.1175/JCLI-D-24-0532.1">https://doi.org/10.1175/JCLI-D-24-0532.1</a>

Van Oldenborgh, G.J., Mitchell-Larson, E., Vecchi, G.A. *et al.* Cold waves are getting milder in the northern midlatitudes. *Environmental Research Letters*, **14(11)**, 114004 (2019). <a href="https://doi.org/10.1088/1748-9326/ab4867">https://doi.org/10.1088/1748-9326/ab4867</a>.

**Response to Reviewer 2**
We gratefully acknowledge the reviewer for his/her remarks and suggestions, which improved the quality of the manuscript significantly. We have carefully considered the suggestions of the reviewer and we provide a point-by-point response to the reviewer's comments. For clarity, the reviewer's comments are given in bold italics and the responses are given in plain text. References that do not refer to those in the main manuscript are listed below. The manuscript will be modified according to the responses that are given to the comments.

**General comments**

This paper addresses temperature and precipitation extremes in the Arctic in CESM experiments with prescibed SST and sea ice. Two versions of CESM, one with uniform resolution and another with a refined grid at high latitudes are considered. First, comparisons with reanalysis and regional model datasets are presented for near-present climate simulations, and second, future changes in extremes are evaluated for two storylines in a high-emission scenario.

The results show that in warmer climate conditions, high temperature extremes and precipitation extremes will become stronger and/or more frequent, while cold/dry extremes will generally become weaker and/or less frequent in the Arctic region. Aside from these somewhat expected findings, there are some differences between the storylines and some (mainly subtler) differences between the experiments performed with the two different grids.

This paper represents an interesting application of the storyline approach to the future Arctic climate, also documenting how the results depend on the model grid. Overall, the paper is thorough and well-written. The technical quality of figures is good, but statistical testing is mostly missing.

We thank the reviewer for his/her evaluation of the manuscript. We have tried to address all concerns.

**Major comments**

1. The evaluation of statistical significance of the findings is largely lacking in this work. In most of the figures, it is not considered at all, and while it is included in some (e.g., Figs. 3 and 13), the method for calculating the statistical significance is not explained. For example, does the statistical testing account for the multiplicity problem? That is, when tests are performed for a large number of grid points, some of them would likely show nominally significant differences even if the results were generated randomly. See, for example:

Wilks, D. S., 2016: "The Stippling Shows Statistically Significant Grid Points": How Research Results are Routinely Overstated and Overinterpreted, and What to Do about It. Bull. Amer. Meteor. Soc., 97, 2263–2273, <a href="https://doi.org/10.1175/BAMSD-15-00267.1">https://doi.org/10.1175/BAMSD-15-00267.1</a>.

Testing the statistical significance is especially relevant because the climate change signal is defined based on relatively short runs (10 years) and because extremes are more subject to the effects of internal variability than mean values.

While, for example, the general increase of warm extremes and wet extremes at most Arctic locations might be robust, it is less obvious how robust are the differences between the changes in extremes for the two storylines, let alone the corresponding differences between POLARRES and NE30 runs. Indeed, it is obvious that some of the figures contain mostly noise (Fig. S13f,h being a particularly "good" example).

I recommend that the statistical significance of the results should be evaluated more systematically, and the reporting should be mainly focused on those aspects found robust. In the case of areal-mean results, such as those Fig. 10 and 12, uncertainty could be illustrated by showing confidence intervals. Just be careful with their interpretation:

Lanzante, J. R., 2005: A Cautionary Note on the Use of Error Bars. J. Climate, 18, 3699–3703, <a href="https://doi.org/10.1175/JCLI3499.1">https://doi.org/10.1175/JCLI3499.1</a>.

Thanks for this valuable comment. For the Figures 3, 6, and 13, we evaluated the statistical significance by using a two-tailed t-test, where outcomes were marked as statistically significant for p-values lower than or equal to 0.05. The two-tailed t-test we performed does not account for the multiplicity problem, which means there is a probability that some points have falsely been marked as statistically significant.

To evaluate the statistical significance in a more systematic way, we will first calculate the extreme metrics on a year-by-year basis before calculating the model-related, grid-related, or time-related differences and its statistical significance. This method deviates slightly from the method presented in the original manuscript as extreme metrics, such as the consecutive dry days index (CDD) and the highest 5-day precipitation sums (RX5day) were based on an entire 10-year period rather than on a year-by-year basis. The statistical significance will be evaluated by performing two-tailed t-tests, except for the areal-mean results as presented in Figures 10 and 12. The p-values generated by the two-tailed t-tests will then be corrected by using the FDR (false discovery rate) approach (Benjamini and Hochberg, 1995; Wilks, 2016) to control for the false positives that emerge from multiple testing. Here we will set  $\alpha_{FDR} = 0.1$  to maintain a global test level of  $\alpha_{alobal} = 0.05$  (Wilks, 2016).

Figure RC1 provides an example of the statistical significance evaluation, showing the annual maximum temperature differences between NE30/POLARRES and ERA5, as well as the differences between POLARRES and ERA5. The stippling here represents the statistical significance of the annual maximum temperature differences at the 95% confidence level. As can be seen in Figure RC1, both NE30 and POLARRES show significant cold temperature differences over northeastern Canada and the central Arctic, as well as significant warm temperature differences over parts of the continental mid-latitudes (Fig. RC1b-c). On average, POLARRES is slightly warmer than NE30 with an average temperature difference of 0.2 °C. Further, POLARRES shows an enhanced cold temperature difference over northeastern Canada, although these differences are not significant. This could be due to the relatively short period over which the differences are examined. Figure RC1 will be part of the revised Figure 5 in the manuscript.

**Figure RC1.** Present-day A (1985–2014) northern hemisphere (poleward of 50°N) annual maximum temperature (TXx (°C)) state (a) and differences between NE30 and ERA5 (b), POLARRES and ERA5 (c), and POLARRES and NE30 (d). The area-weighted root-mean-square-difference (RMSD) and average difference (AVGD) listed above the 2nd-4th column panels are calculated for the domain 60-90°N. The stippling represents the significance of temperature differences at the 95% confidence level.

For the areal-mean outcomes as presented in Figures 10, 12 and S10, we will account for uncertainty ranges by including error bars that show the 95% confidence intervals, which are derived from two-sided t-scores.

Figure RC2 provides an example of the areal-mean outcomes and associated 95% confidence intervals, showing the projected future changes in annual maximum temperature (TXx) for six different regions as defined in the manuscript. The error bars here represent the 95% confidence intervals. As the figure illustrates, all regions show an increase in annual maximum temperature, with the projected changes in extremes being smaller than the projected changes in annual mean near-surface temperature. In addition, the ST2 increases are larger than the ST1 increases, especially over the sea ice region (SIC). Further, the confidence intervals are larger over land, presumably due to the larger heterogeneity in the response of the land surface to future warming. Figure RC2 will be part of the revised Figure 10 in the manuscript.

**Figure RC2.** Projected future (2090–2099) changes in annual maximum temperature (TXx; °C) for six different regions. The bars show the projected changes in NE30 (1st and 3rd bars) and POLARRES (2nd and 4th bars) for ST1 (1st and 2nd bars) and ST2 (3rd and 4th bars). The black dots denote the projected changes in annual mean near-surface temperature (°C). The error bars (black) represent the 95% confidence interval.

To explain the techniques, we will use for the evaluation of the statistical significance and the confidence intervals used for the areal-mean results, we will add an explanation to Section 2.4 *Analysis of temperature and precipitation extremes*. Further we will adjust the main text in the manuscript based on the updated figures.

2. The manuscript is not fully clear about how regridding of the results from different model runs and reanalysis to a common grid is made. For example, is it made using conservative interpolation (which conserves area means) or bilinear interpolation between the nearest grid points, or some other approach? Also, it is not stated clearly whether regridding is applied to the original temperature and precipitation values before calculating the extremes, or whether the extremes are first calculated at the original model resolution and then regridded to the common grid for plotting.

These choices are especially important for precipitation extremes. In particular, I would expect that if (1) bilinear (rather than conservative) interpolation is used and/or (2) the extremes are calculated before regridding, there would be a systematic increase in high precipitation rates (e.g., P99) with improving resolution, simply because daily precipitation values show substantial small-scale variations. On the other hand, if extremes are calculated after a conservative regridding is applied, it is not obvious this will happen (or at least, there would be no trivial reason for that).

There is not necessarily a single "right" solution to how this matter should be handled, but at any rate, you should be clear about how you do it and justify your choice. See also minor comments 11 and 12.

The regridding of the different CESM model output and reanalysis data were performed by bilinear interpolation for temperature and conservative interpolation for precipitation, which are commonly used techniques for regridding temperature and precipitation, respectively. We applied the regridding to the original temperature and precipitation data before the calculation of extremes. To clarify which interpolation techniques we used and when, we will add extra text to Section 2.3 *Model Evaluation* with regard to the interpolation techniques as well as when we applied the regridding:

"Daily temperature fields were regridded using bilinear interpolation and daily precipitation fields were regridded using conservative interpolation. The regridding was applied prior to the analysis of temperature and precipitation climatology and extremes."

**Minor comments**

1. lines 13 and 102: replace the latter "strong/weak" with "weak/strong"? The idea of ST1 and ST2 is to contrast a case with strong land warming and weak SST warming with a case with weak land warming and strong SST warming.

We will replace the latter "strong/weak" with "weak/strong" throughout the manuscript.

2. line 16: It could be mentioned that the better performance of the 1° grid in simulating temperature extremes is related to a larger negative temperature bias in the VR grid. (Incidentally, I think this might be a matter of model tuning rather than any fundamental issue associated with higher resolution).

We will mention this point in the abstract as follows: "The outcomes show that the VR grid generally performs better in simulating precipitation extremes, while the globally uniform 1° grid generally performs better in simulating temperature extremes, which is mainly related to larger negative temperature differences in the VR grid."

3. line 124: You can delete "cloud-aerosol" before "radiation scheme".

We will delete "cloud-aerosol" in the main text.

4. Some parts of the model and experimental description appear unnecessarily detailed in the context of this work. In particular, the description of CLM5 on lines 132–143 and land surface treatment on lines 216-223.

We will shorten the Data and Methods section by shortening the subsections related to the model description (especially the model description of CLM5) as most of the details described in these subsections can also be found in other published work. As for the other parts of the Data and Methods section we will have a thorough look onto which parts of the main text can be shortened.

5. lines 176–177: at least in CICE4, increasing the parameter  $r_{snw}$  actually decreases the snow grain radius over sea ice, and therefore, increases snow albedo. Please check this.

That is indeed correct. The  $r_{snw}$  parameter impacts the snow grain radius. By increasing the parameter  $r_{snw}$  the snow grain radius decreases, which results in a higher snow albedo. We will rephrase the text to clarify the effects of increasing  $r_{snw}$  on the snow grain radius.

**6. lines 182–184: Are microphysical substep and microphysical timestep the same thing or not?**

The microphysical substep ( $micro\_mg\_num\_steps$ ) and the microphysical timestep ( $\Delta t_{mic}$ ) are not the same thing but are related with each other. The microphysical timestep is calculated by dividing the timestep used for both the macrophysics and microphysics ( $\Delta t_{macmic}$ ) by the number of microphysical substeps ( $micro\_mg\_num\_steps$ ):

$$\Delta t_{mic} = \frac{\Delta t_{macmic}}{micro\_mg\_num\_steps}$$

The macrophysics timestep, in turn, is derived from the physics timestep (( $\Delta t_{phys}$ ) by dividing the physics timestep by the number of times the macrophysics and microphysics are subcycled ( $cld\_macmic\_num\_steps$ ):

$$\Delta t_{macmic} = \frac{\Delta t_{phys}}{cld\_macmic\_num\_steps}$$

To give an example, for the NE30 grid  $\Delta t_{phys} = 1800$ s and  $cld\_macmic\_num\_steps = 3$ , which means  $\Delta t_{macmic} = 600$ s. The number of microphysical timesteps (micro\_mg\_num\_steps) equals 8, which means  $\Delta t_{mic} = 75$ s. For the POLARRES grid  $\Delta t_{phys} = 450$  s and  $cld\_macmic\_num\_steps = 2$  (lower for refined grids), which means  $\Delta t_{macmic} = 225$ s. The number of microphysical timesteps ( $micro\_mg\_num\_steps$ ) equals 3, which means  $\Delta t_{mic} = 75$ s. We will rephrase the text to clarify that the microphysical timestep is derived via the number of microphysical substeps.

7. lines 198–203: Please use a notation that it consistent with Levine et al. (2024). Storyline ST1 corresponds to storyline D (not B2) in Levine et al., denoted as ArcAmp+/BKWarm- (not PolAmpl+BKSSTWarm-). Similarly, ST2 corresponds to storyline A (not B1) denoted as ArcAmp-/BKWarm+ in Levine et al.

Thanks for pointing this out. We will revise the notations in the main text.

8. line 213: It would be useful to mention the global-mean near-surface temperature change from the present-day (2005–2014) to the future period (2090–2099) for ST1 and ST2.

We will mention the global mean near-surface temperature change between the future period (2090–2099) and the present-day (2005–2014) for ST1 and ST2 in Section 2.2 *Experimental Design*. The global mean near-surface temperature is projected to increase by 3.3 K for ST1 (represented by the r1i1p1f1 member of NorESM2-MM) and 4.6 K for ST2 (represented by the r1i1p1f2 member of CNRM-ESM2-1).

9. Sect. 2.3. Mention that in regions with sparse observations, such as large parts of the Arctic, the reanalysis fields are strongly influenced by the underlying forecast model.

We will add extra text to Section 2.3 to mention the reanalysis fields can be strongly influenced by the underlying model in areas with little to no observations.

10. line 307–309: If/when CSDI is calculated based on CESM's own climatology, there is no reason why a cold bias would lead to a positive CSDI bias. So the negative CSDI bias is note particularly counter-intuitive.

That is a good point. We will rephrase the text in the following way:

"Cold spell duration index (CSDI) differences are generally negative over much of the central Arctic and Greenland in NE30 and over parts of Scandinavia in POLARRES (Figs. 5n-p), indicating a decrease in the number of cold spell days relative to ERA5. Although the annual mean temperature and the daily minimum temperature (TN) both show a negative temperature difference relative to ERA5 – particularly over the central Arctic and Greenland (Figs. 3 and S1f-h) – the negative cold spell duration differences are not unexpected. A possible explanation for the negative cold spell duration differences is that the variability of the daily minimum temperature differs between ERA5 and CESM. For instance, over sea ice, ERA5 exhibits lower variability in daily minimum temperature compared to CESM (not shown), which may be partly attributed to the fixed sea ice thickness in ERA5 (i.e., 1.5 m; Batrak and Müller, 2019). In CESM, however, the sea ice thickness is variable, which leads to a higher variability in daily minimum temperature and thus a lower probability of meeting the cold spell criteria (Table 1), ultimately resulting in a lower number of cold spell days."

**11. lines 348–357: If the precipitation PDFs are based on daily precipitation rates regridded to the NE30 grid (as stated), then what explains the systematic change in the PDFs (more frequent very high values) with improving horizontal resolution?**

Even though the daily precipitation has been regridded conservatively to the NE30 grid, high extreme precipitation rates associated with high-resolution grids can still shift the tail of the PDF upward and can therefore explain the systematic change in PDFs, showing high extreme precipitation rates more frequently with increasing horizontal resolution.

The increase in the frequency of extreme precipitation rates with increased horizontal resolution can, for instance, be related to 1) better representation of topography, leading to stronger orographic uplifts and orographic precipitation, 2) stronger updrafts, resulting in higher resolved-scale precipitation rates, and 3) an increase in the frequency and intensity of storm systems (e.g., O'Brien et al., 2016; Xu et al., 2022; Zarzycki and Jablonowski, 2014). To clarify what explains the systematic change in PDFs with increased horizonal resolution, we will integrate the above-mentioned explanation into the main text.

With regard to the PDFs shown in Figure 7, we would like to point out that we corrected the figure after discovering an inconsistency in the regridding procedure (i.e. from the source grid to the NE30 grid) prior to calculating the PDFs. This has resulted in a few minor changes to the PDFs.

**12. As a follow-up comment, if Fig. 7 is based on (e.g.) bilinearly interpolated precipitation data, it would be interesting to see how the PDFs behave if conservative interpolation is used instead.**

As mentioned in the former comments, we have used conservative interpolation to regrid precipitation data, which means Fig. 7 is already based on conservatively interpolated precipitation data.

**13. lines 368–369 (also 552–553). Please check if the cited references support your statement regarding extra-tropical storm systems. Based on a cursory reading, they seem to focus on low-latitude systems.**

The work of Xu et al. (2022) focusses on eastern China and the western US and suggests that the greater changes in extreme precipitation events in VR-CESM might be related to a higher frequency and intensity of tropical storm systems, particularly over southern China, which is also supported by the work of Zhu et al. (2023). Although the emphasis of these works is on tropical systems, it still supports the finding that storm systems could increase in frequency and intensity with increased resolution. The work of Zarzycki and Jablonowksi (2014), focusing on tropical storm systems in the Atlantic, found a similar increase in the number of tropical storm systems (including cyclones). As these storm systems can migrate to higher latitudes and can degrade into extra-tropical storm systems, it could be possible that these storms contribute to larger highest 5-day precipitation sum differences over the North American mid-latitudes in POLARRES. As the work of Zarzycki and Jablonowski (2014) has a better potential to support our findings, we have chosen to keep using this citation and to remove the citations of Xu et al. (2022) and Zhu et al. (2023). Further, we will remove "(extra-tropical)" from the main text and rephrase the sentence (lines 368-369) to:

"The larger differences in POLARRES may be partly due to a higher frequency and intensity of storm systems, which bring significant amounts of precipitation over a period of several days (Zarzycki and Jablonowski, 2014)."

And the sentence (lines 531-532) to:

"The larger highest 5-day precipitation sum differences in POLARRES may partly be related to a higher frequency and intensity of storm systems bringing significant amounts of precipitation over a period of several days. However, more research is needed to understand the impact of grid refinement on the frequency and intensity of storm systems in the Arctic as well as its effect on precipitation extremes in CESM."

And the sentence (lines 552-553) to:

"Here, POLARRES projects larger highest 5-day precipitation sum increases, which may partly be related to a higher frequency and intensity of storm systems associated with higher resolution grids."

**14. lines 458–461: It is not clear what these "average" absolute and relative increases represent. The ranges quoted do not cover the actual range of values in Fig. 12. A simple approach would be to take averages over 60-90°N.**

We acknowledge that the listed "average" absolute and relative increases can be confusing. Therefore, we will replace these values with the area averages over 60-90°N.

15. line 547–549: A possible physical interpretation of this is that the regions with high present-day CDD are very high-latitude regions with cold winters, in which daily precipitation above 1 mm is relatively rare (but will increase with warming), while the regions with lower present-day CDD are in the southern parts of Arctic and might experience longer dry periods in summer in a warmer climate.

We agree with the possible physical interpretation of the reviewer. We will integrate this explanation into the main text.

16. line 566: "this narrows the uncertainty range". This requires a more careful reasoning. The definition of storylines in Levine et al. (2024) is based on Arctic land and Barents-Kara Sea warming normalized by global-mean warming. Therefore, while the storyline approach is invaluable for impact studies, it is not obvious to me how it reduces the uncertainty related to the overall Arctic warming.

We agree with the reviewer that the statement "narrows down the uncertainty range" requires a more careful reasoning. We have rephrased this part of the main text as follows:

"The narrower range of projected outcomes suggests that the storyline approach used here may support a more targeted understanding of future Arctic temperature and precipitation extremes, and could help inform adaptation and mitigation strategies in response to climate extremes in the region."

17. Figures 3, 5, 6, 8, S1, S2, S3, S5 and S6: Given that even the renalysis are affected strongly by the underlying model, I suggest to replace root-mean-square error (RMSE) with root-mean-square difference (RMSD), and "Bias" with average difference (AVG or AVGD).

Thanks for the good suggestion. We will change RMSE to RMSD and BIAS to AVGD.

18. Figs. 3 and 6: I suggest to add one more panel showing the difference POLARRES-NE30.

We will add one more panel showing the differences POLARRES-NE30.

19. Fig. 4: Are these PDFs based on daily-mean data? Please mention that in the caption.

The PDFs are indeed based on daily mean temperature data. We will add this information to the figure's caption.

20. Figs. 13 and S15: "... vectors represent the significance". Does this mean that vectors are drawn only where the change is significant (in the third column, they seem to be drawn everywhere!)?

As for the 1st and 2nd columns of Figures 13, S15 and S16, the vectors are indeed only drawn when the changes are statistically significant at the 95% confidence interval. Following a more systematic evaluation of statistical significance as highlighted in general comment 1, we will also include information on the statistical significance in the third columns and will update Figures 13, S15 and S16 accordingly.

**21. Fig. S4: Explain the meaning of stippling.**

The stippling represents the significance of differences at the 95% confidence interval. We will include this information in the caption of Figure S4.

**Language and technical corrections**

1. line 307. Replace "decrease in the number" with "smaller number".

We will replace "decrease in the number" with "smaller number".

2. line 318: Replace "(TXx)" with "and TXx".

We will replace "(TXx)" with "and annual maximum temperature".

3. lines 352–353: "A possible explanation for the lower extreme precipitation ... is that extreme precipitation rates are underestimated". This could be shortened: "It is possible that extreme precipitation rates are underestimated in ERA5 and JRA-3Q".

We will shorten the sentence to "It is possible that extreme precipitation rates are underestimated in ERA5 and JRA-3Q".

4. line 380: "although CESM tends to be slightly drier". You presumably mean "although the dry bias for CESM appears slightly larger"? The CESM results remain unaltered, only the reference changes...

Yes, exactly. We will change the text to "although the dry bias for CESM appears slightly larger"

5. lines 414–415 and 430–432. The use of parentheses to shorten sentences often makes the text more difficult to read. It should be avoided especially in cases in which parentheses are also used in their conventional purpose. See <a href="https://eos.org/opinions/parentheses-are-are-not-for-references-and-clarificationsaving-space">https://eos.org/opinions/parentheses-are-are-not-for-references-and-clarificationsaving-space</a>.

We will have a thorough look at the use of parentheses in the main text and will reduce it where needed.

6. line 419: "response on temperature" should be "effect on temperature"?

We will replace "response on temperature" with "effect on temperature".

6. line 472: replace "wetter and experience" with "wetter but experience"?

We will replace "wetter and experience" with "wetter but experience".

7. line 496: "Fig. 13b-c,e-f". Double-check that these are the correct figure and panel numbers.

Here we refer to Figures 13b-c and 13h-i, which show the future changes in moisture transport convergence, as output by POLARRES for ST1 (Fig. 13b) and ST2 (Fig. 13h), and the differences between POLARRES and NE30 for ST1 (Fig. 13c) and ST2 (Fig. 13i). We will correct the figure/panel numbers to "Fig. 13b-c,h-i"

8. line 519: replace "increased cloud cover" with "overestimated cloud cover".

We will replace "increased cloud cover" with "overestimated cloud cover".

9. caption of Fig. 1: SIC and OCN are marked with colours rather than diagonal and crossed pattern.

SIC is marked with cyan and OCN is marked with blue. We will correct the description in the caption of Fig. 1.

10. Fig. S7: "Same as Figs. S8 and S9". It is not reader-friendly to refer to later figures in the caption. Move Fig. S7 after S8 and S9?

We will move Fig. S7 after S8 and S9 and will adjust the figure numbers and main text accordingly.

**References**

Benjamini, Y. and Hochberg, Y. Controlling the false discovery rate: a practical and powerful approach to multiple testing. *Journal of the Royal Statistical Society: Series B (Methodological)*, **57(1)**, 289-300 (1995). https://doi.org/10.1111/j.2517-6161.1995.tb02031.x

Wilks, D.S. "The Stippling Shows Statistically Significant Grid Points": How Research Results are Routinely Overstated and Overinterpreted, and What to Do about It. *Bull. Amer. Meteor. Soc.*, **97**, 2263–2273 (2016). <a href="https://doi.org/10.1175/BAMSD-15-00267.1">https://doi.org/10.1175/BAMSD-15-00267.1</a>.